# A prostate-specific membrane antigen activated molecular rotor for real-time fluorescence imaging

Jingming Zhang[1,13], Anastasia Rakhimbekova[2,13], Xiaojiang Duan[1,13], Qingqing Yin[3], Catherine A. Foss[4], Yan Fan[1], Yangyang Xu[5,6,7,8], Xuesong Li[5,6,7,8], Xuekang Cai[1], Zsofia Kutil[2], Pengyuan Wang[9], Zhi Yang[10], Ning Zhang [11], Martin G. Pomper[4], Yiguang Wang [3 ✉], Cyril Bařinka [2 ✉] & Xing Yang [1,12 ✉]

Surgery is an efficient way to treat localized prostate cancer (PCa), however, it is challenging to demarcate rapidly and accurately the tumor boundary intraoperatively, as existing tumor detection methods are seldom performed in real-time. To overcome those limitations, we develop a fluorescent molecular rotor that specifically targets the prostate-specific membrane antigen (PSMA), an established marker for PCa. The probes have picomolar affinity ($IC_{50} = 63\text{-}118$ pM) for PSMA and generate virtually instantaneous onset of robust fluorescent signal proportional to the concentration of the PSMA-probe complex. In vitro and ex vivo experiments using PCa cell lines and clinical samples, respectively, indicate the utility of the probe for biomedical applications, including real-time monitoring of endocytosis and tumor staging. Experiments performed in a PCa xenograft model reveal suitability of the probe for imaging applications in vivo.

[1] Department of Nuclear Medicine, Peking University First Hospital, 100034 Beijing, China. [2] Institute of Biotechnology of the Czech Academy of Sciences, BIOCEV, Prumyslova 595, 25250 Vestec, Czech Republic. [3] State Key Laboratory of Natural and Biomimetic Drugs, Peking University, 100191 Beijing, China. [4] Russell H. Morgan Department of Radiology and Radiological Science, Johns Hopkins University School of Medicine, Baltimore, MD 21205, USA. [5] Department of Urology, Peking University First Hospital, 100034 Beijing, China. [6] The Institute of Urology, Peking University, 100034 Beijing, China. [7] National Urological Cancer Center, 100034 Beijing, China. [8] Beijing Key Laboratory of Urogenital Diseases (Male) Molecular Diagnosis and Treatment Center, 10034 Beijing, China. [9] Department of General Surgery, Peking University First Hospital, 100034 Beijing, China. [10] Key Laboratory of Carcinogenesis and Translational Research (Ministry of Education/ Beijing), Department of Nuclear Medicine, Peking University Cancer Hospital & Institute, 100142 Beijing, China. [11] Translational Cancer Research Center, Peking University First Hospital, 100034 Beijing, China. [12] Institute of Medical Technology, Peking University Health Science Center, 100191 Beijing, China. [13]These authors contributed equally: Jingming Zhang, Anastasia Rakhimbekova, Xiaojiang Duan. ✉email: yiguang.wang@pku.edu.cn; cyril.barinka@ibt.cas.cz; yangxing2017@bjmu.edu.cn

Prostate cancer (PCa) is the second leading cause of cancer-related death in men. Patients with localized tumors can benefit greatly from radical prostatectomy[1,2]. Long-term control can be achieved if the tumor-positive margin is completely resected. However, that can be challenging unless there is a reliable way to visualize very small amounts of tumor that may still reside in the margin after resection. Currently, histopathology is the most reliable way to identify a positive margin, but it is time-consuming, and is not often uncovered until the postoperative period. An imaging agent that could reliably identify a positive margin intraoperatively may change the surgical plan in real-time and contribute to a better outcome.

Prostate-specific membrane antigen (PSMA) is a type II transmembrane glycoprotein specifically overexpressed in most PCa with limited expression in normal tissue[3,4]. It is regarded as a high-value biomarker for PCa and has received worldwide attention for PCa diagnosis and treatment. PSMA-targeting ligands have succeeded in guiding a variety of functional groups for such purposes, including radionuclides, fluorophores, photosensitizers, Gd (III) and so on[5–12]. With the development of high-quality near-infrared dyes, PSMA-targeted fluorescent ligands are being actively pursued[5,13]. Upon administration, a PSMA-targeted fluorescent dye can specifically retain within PCa, and after clearance of background signal from the blood pool, which normally takes several hours, PSMA-specific tumor imaging can be achieved, and may enable surgical guidance. To date optical probes targeting prostate cancer have largely been based on fluorescent dyes (Cy7, IRdye800CW, indocyanine green etc.) that are "always on", requiring substantial time to achieve suitable image contrast[14]. PSMA-activatable fluorescent probes, which have an off-on response, will be desirable as they may have less background than fluorescent agents that are fluorescent upon excitation, but only a few examples have been reported[15–17]. Kobayashi et al. carried out pioneering studies based on the quenching effect when a fluorescent dye (such as indocyanine green) was conjugated with an antibody or minibody, demonstrating a 30-fold increase in fluorescence upon activation and internalization of the agent by PSMA[15,16]. Based on that activation effect, they could detect PSMA-positive tumors specifically at 6 h post-injection with minibodies conjugated with indocyanine green. However, the requirement of internalization for activation, and the long clearance times of monoclonal antibodies from the blood pool, prevent imaging shortly after administration. Urano et al. recently reported a novel fluorogenic method using the carboxypeptidase activity of PSMA. The method could be utilized for ex vivo fluorescence imaging of PCa in surgically resected clinical specimens[17]. The method enables high fluorescence enhancement, while the generation of signal depends on the enzymatic activity of PSMA, and requires minimum 30 min for sample staining. Overall, existing methods need at least tens of minutes to hours to reach suitable image contrast, encouraging us to develop probes that may enable imaging at shorter time post-administration.

Fluorescent molecular rotors (FMRs) are a family of fluorophores sensitive to local microenvironment (e.g., polarity and viscosity)[18–20]. Upon photoexcitation, the molecule can form a low-energy state, referred to as a twisted intramolecular charge transfer (TICT) state, so that the excitation energy can be dissipated accompanied by red-shifted emission or non-radiative relaxation. The strategy for applying FMRs is to restrict the formation of the TICT state, in which case specific fluorescence enhancement (usually quantum yield and fluorescent lifetime) can be obtained[21]. The fluorescence response of FMRs is more sensitive and faster compared with other on-off probes mediated by specific chemical reactions[22], enabling real-time and in situ detection. In this regard, some FMRs have been developed for sensing viscosity in the microenvironment, such as derivatives of julolidine (DCVJ, CCVJ)[23,24], meso-phenyl-substituted derivatives of BODIPY[25–27], porphyrin derivatives[28], and merocyanine dyes[29]. Increasingly, FMRs have been exploited for imaging biomolecules (protein, DNA, RNA, peptidoglycan) and biomolecular interactions[30–38]. We hypothesize that by conjugating to a low-molecular-weight PSMA inhibitor, the FMR can interact with residues lining the entrance funnel of the enzyme, whereupon activation will occur[12,39] (Fig. 1a). Since that process is triggered simply by binding to PSMA and does not involve other steps, such as endocytosis and catalysis, it will be expected to respond rapidly and specifically. Here we report the discovery of FMR-based probes specifically activated by PSMA. We reveal the underlying mechanism of the fluorescent activation upon probe-PSMA binding. Furthermore, these probes are successfully applied to real-time monitoring of PSMA-mediated endocytosis, rapid prostate cancer tissue staining, and in vivo imaging of PCa.

## Results

### Design and synthesis of PSMA-targeted activatable probes.
FMRs based on benzonitrile and julolidine moieties are well known[24]. Recently, these have been successfully modified by Yen-Pang Hsu et al. and FMRs with wavelengths closer to the near-infrared region ($\lambda_{ex}/\lambda_{em} = 490/660$ nm) have been generated. Those modifications enabled better tissue penetration of a fluorescence signal for imaging applications in vitro or in vivo[37]. In this study, we aimed at the design of activatable probes by linking a benzonitrile fluorescent rotor to the Lys-Urea-Glu scaffold, a well-established PSMA-targeting moiety used clinically for PCa imaging and radiotherapy[39]. To maximize critical interactions between the benzonitrile fluorescent rotor and the protein upon ligand binding, the fluorophore was directly conjugated to the Lys-Urea-Glu scaffold at the ε-amine of lysine to yield Glu-490 (Supplementary Fig. 1). Additionally, we synthesized ODAP-490 and ODAP-436, Lys-Urea-oxalyldiaminopropionic acid analogs of the traditional glutamate-containing scaffold, to increase its water solubility and reduce nonspecific interaction. (Fig. 1b)[12].

### Biochemical and biophysical characterization of probes.
To evaluate fluorescence enhancement of our FMR probes in relation to their molecular environment, initially, we used a glycerol solution to mimic rotationally constrained conditions likely found upon PSMA binding and collected fluorescence and UV-Vis spectra of the compounds[37]. ODAP-436 exhibited UV absorption and fluorescence emission maxima at 436 and 490 nm, respectively. Both Glu-490 and ODAP-490 exhibited UV absorption and fluorescence emission maxima at 490 and 660 nm, respectively (Fig. 1b). All probes showed a marked glycerol-dependent fluorescence increase, with an over 27-fold signal enhancement at 80% glycerol concentration compared to pure buffer (Fig. 1c).

Using recombinant human PSMA (rhPSMA), we next determined critical biophysical and biochemical parameters of the probes upon complexation with the enzyme in solution. First, $IC_{50}$ values of the probes were determined using an established NAAG-hydrolyzing assay[40]. Inhibition constants for Glu-490, ODAP-490, and ODAP-436 were 63.1, 99.6, and 118.4 pM (Fig. 1d), respectively, identifying these compounds as potent PSMA-specific inhibitors. Next, we evaluated the stoichiometry and fluorescence enhancement of probes upon rhPSMA binding. To that end, probes at a concentration of 500 nM were titrated with a dilution series of rhPSMA revealing the expected 1:1 stoichiometry of binding (Fig. 1e). Furthermore, we observed a 29.9- to 38.3-fold increase in fluorescence intensity (Fig. 1f), pointing towards efficient rotational constraints imposed on the fluorophore upon PSMA binding. Finally, we evaluated the time

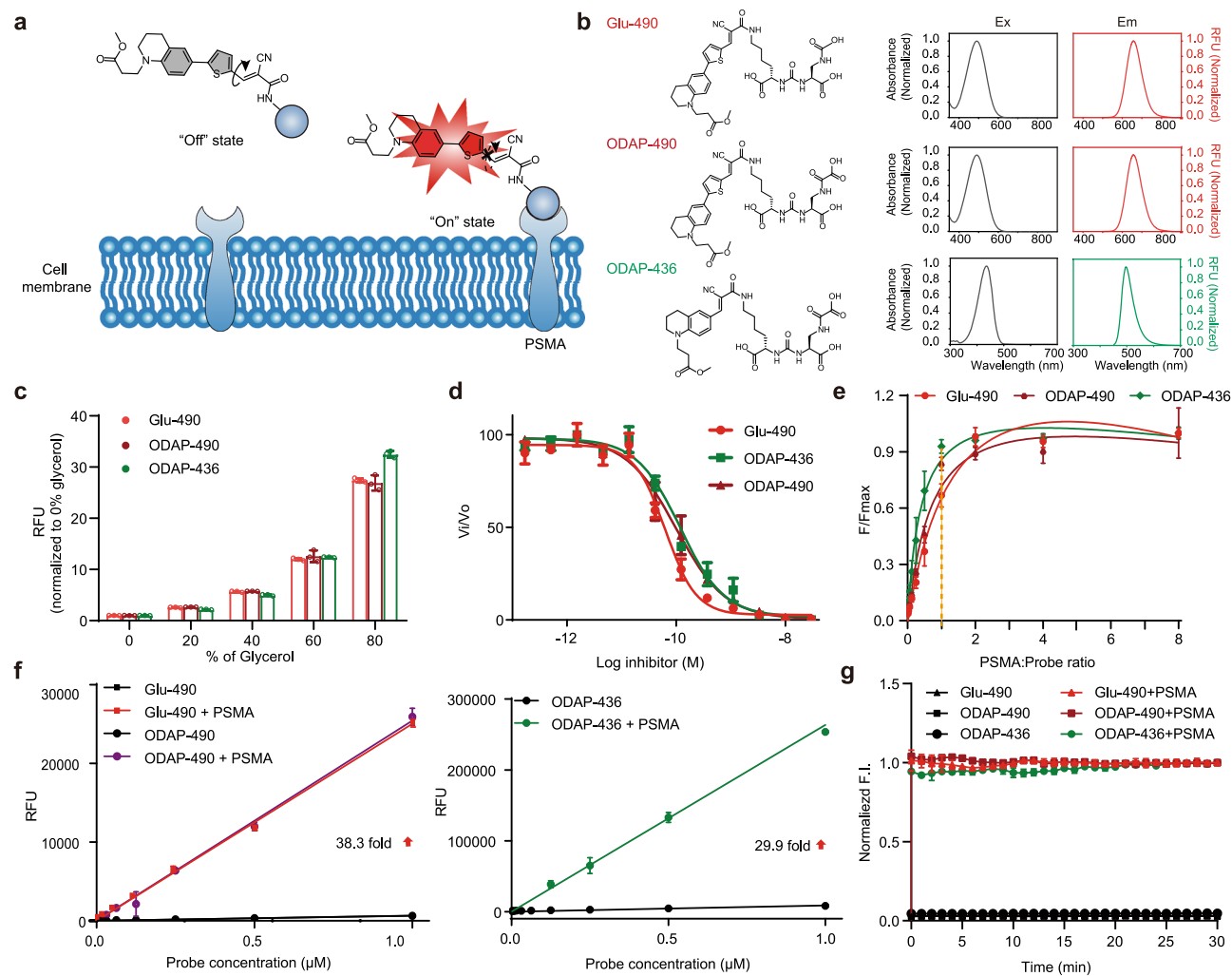

**Fig. 1 Strategy, structures, and fluorescence properties of the PSMA-activatable probes. a** Schematic of probe binding and activation. **b** Structures, absorbance, and fluorescence spectra of PSMA activated fluorescent probes: Glu-490, ODAP-490, and ODAP-436. RFU relative fluorescence units. **c** Changes in fluorescence intensity of probes in solutions with increasing glycerol concentrations. Probe concentration was 0.1 mM and data were normalized to PBS control. Data are presented as mean values ± s.d. ($n = 3$ biologically independent experiments). **d** Inhibition of PSMA enzymatic activity using the radioenzymatic assay. Data are presented as mean values ± s.d. ($n = 2$ biologically independent experiments). **e** Saturation binding of rhPSMA/ probe complexes. Data are presented as mean values ± s.d. ($n = 3$ biologically independent experiments). **f** Fluorescence intensity of the rhPSMA/probe complex in response to concentration changes. Data are presented as mean values ± s.d. ($n = 3$ biologically independent experiments). **g** Time frame for rhPSMA/probe complex formation. Data are presented as mean values ± s.d. ($n = 3$ biologically independent experiments). F.I. fluorescence intensity. Source data are provided in Source Data file.

required for PSMA activation of the probe. Here, a probe at 500 nM concentration was mixed with a 10-fold molar excess of PSMA, and the fluorescence signal measured over a 30 min time interval (Fig. 1g). The fluorescence signal reached its maximum in less than 30 sec, the time necessary for mixing the components and start of the measurement, and remained stable during the entire experiment.

Overall, compounds Glu-490, ODAP-490, and ODAP-436 provide nearly instantaneous and robust, stable fluorescence upon rhPSMA binding. The combination of these characteristics makes these probes ideal for biological and biomedical applications where time resolution is critical.

**Mechanism of fluorescent activation upon probe-PSMA binding.** To provide mechanistic rationale for the fluorescent enhancement of Glu-490 upon rhPSMA binding, we solved the crystal structure of the rhPSMA/Glu-490 complex and refined it to the 1.73 Å

resolution limit (Table 1). The active-site-bound Glu-490 was fitted into well-defined interpretable *Fo-Fc* electron density peaks in the final stages of the refinement (Fig. 2b). Positioning of the Lys-Urea-Glu motif is virtually indistinguishable from rhPSMA/urea ligand complexes reported previously[41]. The P1' glutamate moiety binds the S1' pocket in the "canonical" mode, the urea linker engages several residues in the vicinity of the active-site zinc ion, and at the nonprime side, the most noticeable contacts involve the arginine patch of PSMA and the P1 α-carboxylate function of Glu-490 lysine function[42,43].

The FMR moiety consists of a distal N-substituted tetrahydroquinoline donor group, a thiophene ring spacer and a nitrile acceptor group (Fig. 2a). In our structure, the three subgroups adopt a near-planar configuration (Fig. 2b, c) that is sustained via intermolecular interactions with PSMA residues. That spatial arrangement prevents the intramolecular twisting motion of the subgroups relative to each other and such rigidity is critical for efficient fluorescence emission upon FMR photoexcitation[44]. The

**Table 1 Data collection and refinement statistics.**

|  | PSMA/Glu-490 |
|---|---|
| *Data collection* | |
| Space group | I222 |
| Cell dimensions | |
| *a, b, c* (Å) | 101.54, 130.10, 158.91 |
| α, β, γ (°) | 90, 90, 90 |
| Resolution (Å) | 50–1.73 (1.83–1.73)[a] |
| $R_{merge}$ | 0.062 (0.564) |
| $I / \sigma I$ | 14.43 (1.99) |
| Completeness (%) | 99.6 (98.9) |
| Redundancy | 4.5 (4.2) |
| *Refinement* | |
| Resolution (Å) | 46.96–1.73 (1.77–1.73) |
| No. reflections | 106,994 (7848) |
| $R_{work} / R_{free}$ | 16.3/18.3 (28.4/28.0) |
| No. atoms | 6622 |
| Protein | 5836 |
| Ligand/ion | 1/4 |
| Water | 468 |
| *B*-factors | 36.5 |
| Protein | 34.8 |
| Ligand/ion | 42.4 |
| Water | 40.6 |
| R.m.s. deviations | |
| Bond lengths (Å) | 0.017 |
| Bond angles (°) | 1.71 |

[a]Dataset was collected from a single crystal. [a]Values in parentheses are for highest-resolution shell.

nitrile group is inserted into a deep pocket delineated the main chains of G206–F209 and the Y700 side chain. Furthermore, the FMR ring system is propped against the PSMA surface delineated by residues N698–G702 of the "glutarate sensor" (Fig. 2d)[45]. Here, the most conspicuous are CH–π interactions between the thiophene ring and the Y700 methylene group (4.0 Å) and the tetrahydroquinoline moiety and the A701 side chain methyl group (3.8 Å and 4.1 Å; Fig. 2c).

It can be surmised that the FRM moiety also comes intermittently into a contact with residues F546 and S547 of the entrance lid, a flexible segment at the entrance into the internal PSMA cavity (amino acids Y541–G548)[42]. However, the weaker and discontinuous *Fo-Fc* electron density map prevented us from generating this part of the model with high confidence (Supplementary Fig. 6).

**Real-time monitoring of live cell receptor binding and endocytosis.** As Glu-490 and ODAP-490 were virtually indistinguishable in our biophysical experiments, and ODAP-490 exhibited less nonspecific staining than Glu-490 (Supplementary Fig. 8), we selected ODAP-490 for the ensuing biological applications. Using LNCaP, 22RV1, and PC3 cell lines expressing high, medium-to-low, and negligible levels of PSMA[46,47], respectively, we first evaluated the cytotoxicity of ODAP-490 using an established MTT-based viability assay[48], the probe proved non-toxic at concentrations up to 25 μM (Fig. 3a, Supplementary Fig. 7). For imaging of PSMA-expressing cells, cells were incubated with 10 μM ODAP-490 for 2 h and subsequently visualized by fluorescence microscopy (Fig. 3b). Quantification of the fluorescent signal revealed correlation between PSMA expression levels and fluorescence intensity with LNCaP and 22RV1 showing a 4.4- and 2.3-fold higher signal compared to PC3 (Fig. 3c).

To monitor PSMA endocytosis in real-time, LNCaP cells were treated with 10 μM ODAP-490 and cells were imaged using a confocal microscope at 37 °C for 2 h in 3 min intervals (Fig. 3e,

Supplementary Movie 1). Fluorescence intensity rose rapidly within the first 6 min and then the signal increased gradually for the remaining time, plateauing at around 90 min. The time profile likely reflects rapid binding of the probe to the plasma membrane-resident PSMA molecules within the first 6 min followed by continuous PSMA internalization and recycling back to the plasma membrane (Fig. 3g). The specificity of fluorescence monitoring was verified using ZJ-43, an established, potent PSMA inhibitor[49]. By pretreating LNCaP cells with 100 μM ZJ-43, the fluorescence signal was significantly reduced (15 fold), consistent with competition between the probe and ZJ-43 for PSMA binding (Fig. 3f, g, Supplementary Movie 2).

Several known modulators of endocytosis were then used to evaluate the mechanism of PSMA trafficking. As expected, the endocytosis was blocked nearly completely by incubating cells at 4 °C, yet the strong fluorescence signal was still observed at the cell surface (Fig. 3d). Chlorpromazine (CPZ), nystatin, and 5-(N-ethyl-N-isopropyl) amiloride (EIPA) were used to target clathrin-dependent[50], caveolin-dependent[51], and macropinocytotic[52] pathways, respectively. Cells were pretreated with 10 μM concentration of each inhibitor for 30 min prior to addition of ODAP-490, and the fluorescence signal was monitored. CPZ exhibited the most significant, ~86%, inhibition of the PSMA-mediated ODAP-490 uptake, conforming a major role for the clathrin-dependent pathway PSMA endocytosis (Supplementary Fig. 9, Supplementary Movie 3).

**Fast and convenient staining of PCa tissues ex vivo.** Human histological samples were used ex vivo to evaluate the feasibility of monitoring PSMA expression levels by ODAP-490. We first co-stained frozen sections of PCa surgical specimens using an anti-PSMA antibody and ODAP-490 (Fig. 4a), and observed correlation between the green fluorescence of antibody and the red fluorescence of ODAP-490 with a Pearson's coefficient 0.88 (Fig. 4b).

Six prostatectomy specimens were harvested and stained by ODAP-490. Differences in fluorescence intensity were observed in focal regions of each sample, likely reflecting heterogeneity of PSMA expression (Fig. 4c). For each sample, three regions of interest were selected, excised, and examined by a pathologist for PSMA expression and Gleason score, which correlated with a Pearson's coefficient of 0.69 (Fig. 4c, d, e). Without additional interventions, such as fixing or washing, the method enabled a convenient way to identify PCa rapidly in patient specimens.

**In vivo imaging of PCa.** Coupled with the fluorescence switching characteristics of ODAP-490, we speculated that this probe can detect PCa in vivo at early timepoints. We tested the ability of ODAP-490 to detect PCa after intravenous injection into mice bearing PSMA-expressing prostate tumors. 22RV1 tumors with medium-to-low level of PSMA expression were selected to challenge the detection limit, and PC3 tumors were used as a negative control. After administration of 25 nmol ODAP-490, we found the tumor could be clearly visualized at 2 h after injection, with the combination effect of specific signal activation inside the tumor and the clearance of the background signal from blood pool (Fig. 5a, b). At the same condition, there was no obvious fluorescence signal accumulation in PC3 xenografts. This result indicates that ODAP-490 is suitable for specifically in vivo imaging of PSMA-expressing tumor.

We next compared the in vivo imaging capability of PCa of ODAP-490 with an established "Always-On" PSMA-targeting probe ODAP-800CW[12]. Mice were administrated with ODAP-490 or ODAP-800CW, and the images at 4 h post-injection were collected. With the tumor specific activatable property,

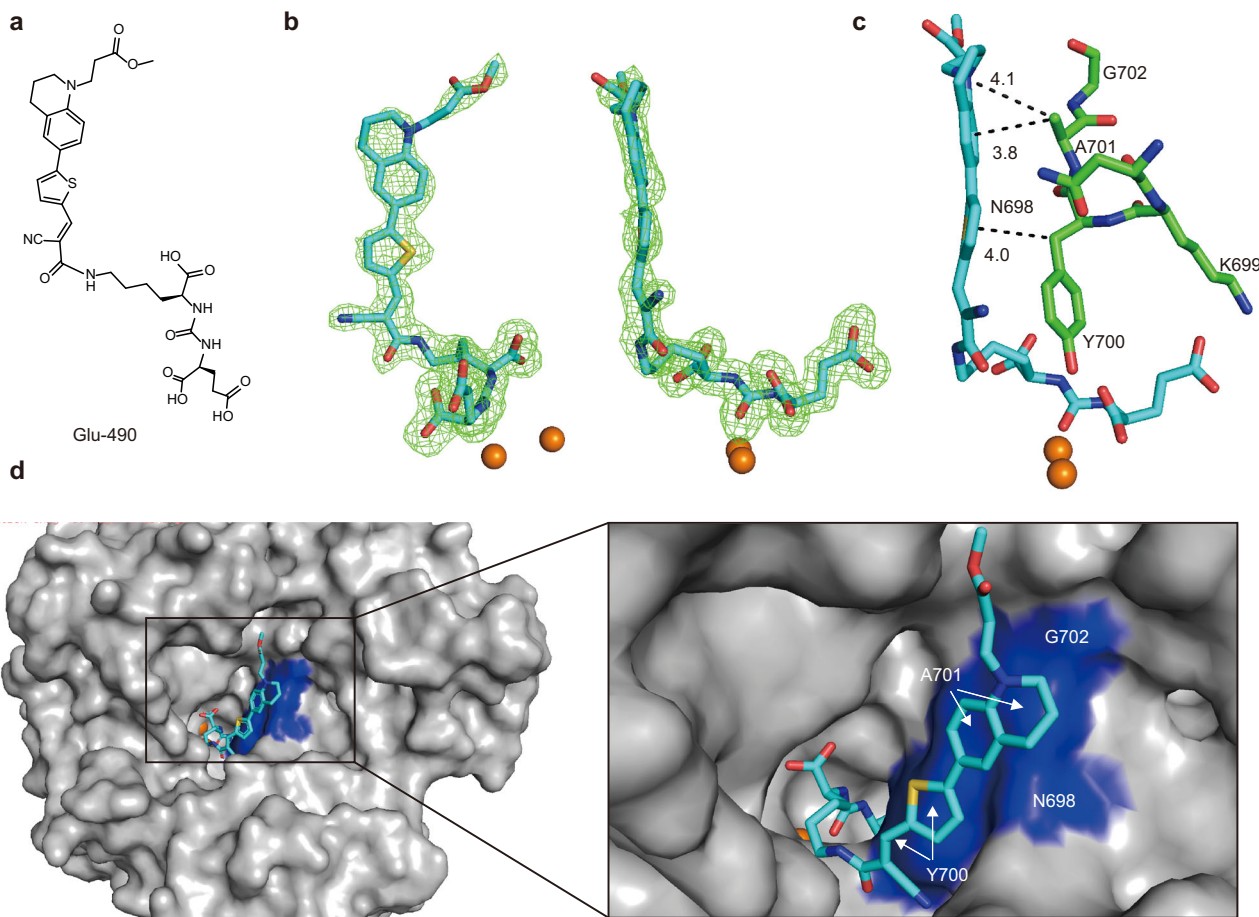

**Fig. 2 Structural characterization of the PSMA/Glu-490 complex. a** Molecular formula of the Glu-490. **b** A stereo view of the Gluo-490 inhibitor. The *Fo-Fc* omit map (green) is contoured at 3.0 σ and the inhibitor is shown in stick representation with atoms colored red (oxygen), blue (nitrogen), yellow (sulfur), and cyan (carbon). **c** Details of interactions between residues of the glutarate sensor (green carbons) and Glu-490 (cyan carbons). CH–π interactions are depicted as dashed lines with distances to the ring centers in Angstroms. The active-site zinc ions are shown as orange spheres. **d** Surface representation of PSMA with residues of the glutarate sensor interaction with the FMR moiety colored blue, PDB code (7BFZ).

ODAP-490 could reach a high tumor-to-background ratio of 29.1 folds, in comparison with ODAP-800CW barely reaching 2.4 folds (Fig. 5c, d). These preliminary in vivo results demonstrated the potential advantages of PSMA-activatable probe for faster tumor detection, although the wavelength of the activatable dye still need further effort to improve.

## Discussion

PSMA, specifically overexpressed in nearly all PCa with limited expression in normal tissues, is an ideal biomarker for imaging PCa with high sensitivity and specificity. To overcome the time-consuming limit of fluorescent PSMA detection, we designed a series of PSMA-activatable fluorescence probes by facile conjugation of FMRs to a PSMA-targeting moiety. The probes bind to PSMA with very high potency (63–118 pM). We showed that upon PSMA binding, the FMR tightly fitted into the entrance funnel of the enzyme restricting conformation of the fluorophore (PDB code: 7BFZ), and therefore eliciting the fluorescent signal (29.9 to 38.3 fold). Since that process is triggered directly and only by PSMA/probe complex formation, the instant signal turning-on can be achieved within 30 seconds, which represents the fastest PSMA activated probes discovered so far. The high PSMA binding affinity and fast signal response allowed us to test a series of biomedical applications.

The PSMA expression on live cells can be visualized using a convenient wash-free protocol by simply incubating with the probe. With this protocol, we were able to monitor and quantify the entire PSMA/probe complex formation and endocytosis process in living cells in real-time. By quantifying the effects of different types of endocytosis inhibitors on PSMA-mediated endocytosis, we confirmed that the clathrin-dependent endocytosis pathway plays the most important role in the endocytosis of PSMA[53,54], while the contribution of the caveolin-dependent and micropinocytosis pathway is quite limited. That is consistent with the established results that most transmembrane receptors are internalized via a clathrin-dependent mechanism, as recently shown by Matthias et al., who did so using stimulated emission depletion (STED) nanoscopy[55].

Meanwhile, a similar wash-free protocol could be adapted for staining PCa surgical specimens and we were able to achieve a quick and accurate PSMA quantification within 10 min, correlating with cancer existence and Gleason score. Compared with conventional time-consuming histopathology, the wash-free method could potentially be adapted intraoperatively. Furthermore, the specific in vivo imaging of ODAP-490 could be achieved in mouse xenografts within a few hours. Compared with the "Always-On" PSMA-targeted dye ODAP-800CW[12], ODAP-490 achieved a significant improved tumor-to-background contrast at the early timepoints, making it a special candidate for intraoperative surgical guidance when imaging time is critical.

In summary, we have developed a series of PSMA-activatable fluorescence probes, which could exhibit a fast fluorescence

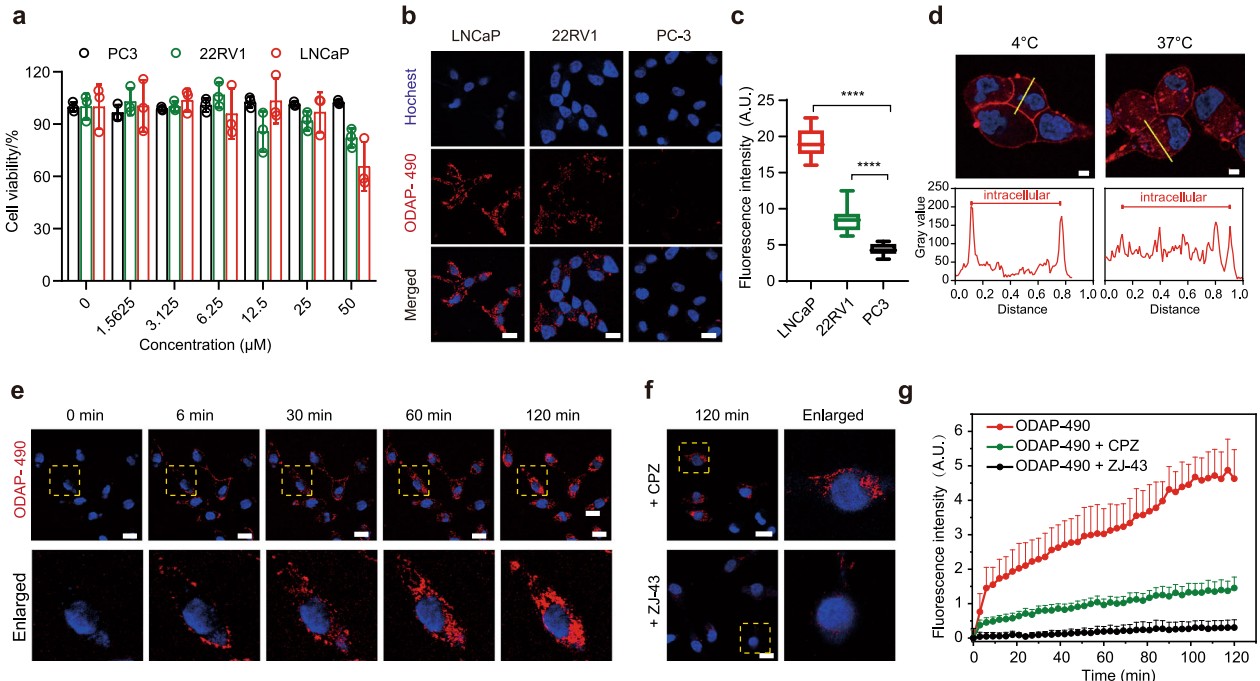

**Fig. 3 Wash-free imaging of PSMA and real-time imaging of PSMA-mediated endocytosis. a** Cytotoxicity of ODAP-490 determined by the MTT viability assay. Data are presented as mean ± s.d. (n = 3 independent experiments). **b** Wash-free imaging of LNCaP (PSMA+++), 22RV1 (PSMA+), and PC3 (PSMA−) cells. 10 μM ODAP-490 was applied at 37 °C for 2 h. Cell nuclei were counter-stained with hochest 33342. (red, ODAP-490; blue, hochest 33342). Scale bar: 20 μm. **c** Quantification of the fluorescence intensity of samples in panel **b**. Upper and lower bounds of boxes represent 25th and 75th percentiles, horizontal lines indicate the median values, whiskers represent the minimum and maximum ranges. (n = 20 biologically independent cell samples). ****$P < 0.0001$, P values = $6.35e^{-22}$(LNCaP); $2.90e^{-11}$(22RV1); two-tailed unpaired Student t-test. A.U. arbitrary units. **d** Inhibition of PSMA-mediated endocytosis at 4 °C. The bottom figures show the fluorescence intensities corresponding to cross-sections (yellow lines) in the upper figures. (red, ODAP-490; blue, hochest 33342). Scale bar: 5 μm. Experiment was repeated three times independently with similar results. **e** The real-time monitoring of PSMA ligand binding and endocytosis. LNCaP cells were treated with 10 μM ODAP-490 and images were captured with a confocal microscope at 37 °C for 2 h at 3 min intervals. (red, ODAP-490; blue, hochest 33342). Scale bar: 20 μm. **f** Inhibition of the PSMA binding and PSMA-mediated endocytosis. The 2 h time point images are shown. (red, ODAP-490; blue, hochest 33342). CPZ chlorpromazine. Scale bar: 20 μm. **g** Quantification of fluorescence intensity changes of endocytosis with/without inhibitors in panels **e** and **f**. Data are presented as mean + s.d. (n = 20 biologically independent cell samples). Source data are provided in Source Data file.

intensity increase upon binding to PSMA. The probe was successfully applied to real-time monitor the whole process of PSMA ligand binding and mediated endocytosis in live cells, and quickly wash-free stain PCa surgical specimen. In vivo imaging of PCa was also successfully demonstrated with the advantage for detecting early timepoints. Effort to further improve the wavelength of PSMA-activatable probes is currently under investigation.

## Methods

**General information**. All solvents and chemicals were purchased from commercial sources, with purity of analytical grade or better, and were used without further purification. Analytical thin-layer chromatography (TLC) was performed using Merck aluminum-backed silica gel 60 F254 (Billerica, MA). Preparative column chromatography was performed on a Bonna-Agela Technologies Co., Ltd. FL-H050G preparative chromatography system (Tianjin, China) equipped with a Phenomenex C18 Luna 10.0 × 250 mm² column. The products were eluted by mixing eluent A (water with 0.1% trifluoroacetic acid) and eluent B (acetonitrile with 0.1% trifluoroacetic acid) with different ratios. NMR spectra were recorded on a Bruker 400 or 600 MHz spectrometer and chemical shifts (δ) were reported in ppm using solvent residual peak as an interior label. High-resolution ESI mass spectra were obtained on an Agilent 6545 triple quadrupole LC − MS instrument (Santa Clara, CA). The characterization data of all compounds are provided in the Supplementary Information.

## Synthesis of PSMA-activatable probes

*(((S)-1-carboxy-5-(2-cyano-3-(5-(1-(3-methoxy-3-oxopropyl)-1,2,3,4-tetrahydroquinolin-6-yl)thiophen-2-yl)acrylamido)pentyl)carbamoyl)-L-glutamic acid (Glu-490).* 2-cyano-3-(5-(1-(3-methoxy-3-oxopropyl)-1,2,3,4-tetrahydroquinolin-6-yl)thiophen-2-yl)acrylic acid (1) (100 mg, 0.25 mmol)[37], 1-[Bis(dimethylamino)methylene]-1H-1,2,3-triazolo[4,5-b]pyridinium 3-oxid hexafluorophosphate (HATU) (191 mg, 0.50 mmol), and

N-ethyldiisopropylamine (DIPEA, 324 mg, 2.51 mmol) were mixed and stirred in methylene chloride (10 mL) for 10 min at room temperature. Then di-tert-butyl (((S)-6-amino-1-(tert-butoxy)-1-oxohexan-2-yl)carbamoyl)-L-glutamate (2a) (246 mg, 0.50 mmol)[56] was added. The reaction mixture was stirred for 2 h. After completion of the reaction, the mixture was diluted with methylene chloride (100 mL) and washed with H₂O twice and brine once. The organic layer was dried over anhydrous Na₂SO₄. After the solvent was removed, the crude material was dissolved in trifluoroacetic acid/methylene chloride (1:1, 10 mL) and stirred at room temperature for 3 h. The product was purified by reverse phase HPLC (0–5 min, 10% MeCN (0.1%TFA); 5–15 min, 10–70% MeCN(0.1%TFA); 15–25 min, 70% MeCN (0.1%TFA); R_t = 20.1 min) to yield Glu-490 as a red solid (18 mg, yield 10%).

*(4 S,8 S)-15-cyano-16-(5-(1-(3-methoxy-3-oxopropyl)-1,2,3,4-tetrahydroquinolin-6-yl)thiophen-2-yl)-1,6,14-trioxo-2,5,7,13-tetraazahexadec-15-ene-1,4,8-tricarboxylic acid (ODAP-490).* Compound 1 (300 mg, 0.76 mmol), HATU (288 mg, 0.76 mmol), and DIPEA (100 μL) were mixed and stirred in methylene chloride (10 mL) for 10 min at room temperature. Then, tert-butyl (((S)-1-(tert-butoxy)-3-(2-(tert-butoxy)-2-oxoacetamido)-1-oxopropan-2-yl)carbamoyl)-L-lysinate (2b) (280 mg, 0.55 mmol)[12] was added. The reaction mixture was stirred for 2 h. After the reaction completed, the mixture was diluted with methylene chloride (100 mL) and washed with H₂O twice and brine once. The organic layer was dried over anhydrous Na₂SO₄. After the solvent was removed, the crude material was dissolved in trifluoroacetic acid/methylene chloride (1:1, 10 mL) and stirred at room temperature for 3 h. The crude product was purified by reverse phase HPLC (0–3 min, 10% MeCN (0.1%TFA); 3–18 min, 10–70% MeCN (0.1%TFA); 18–25 min, 90% MeCN (0.1%TFA); R_t = 16.0 min) to yield ODAP-490 as a red solid (50 mg, yield 12.4%).

*Methyl-3-(6-formyl-3,4-dihydroquinolin-1(2H)-yl)propanoate (5).* Compound 4 (4.8 g, 21.8 mmol), dimethylformamide (16.9 mL, 217.9 mmol) and dichloromethane (DCM, 44 mL) were added to a 250 mL round bottom flask. The reaction mixture was stirred under 0 °C. Phosphoryl chloride (4.1 mL, 42.0 mmol) was

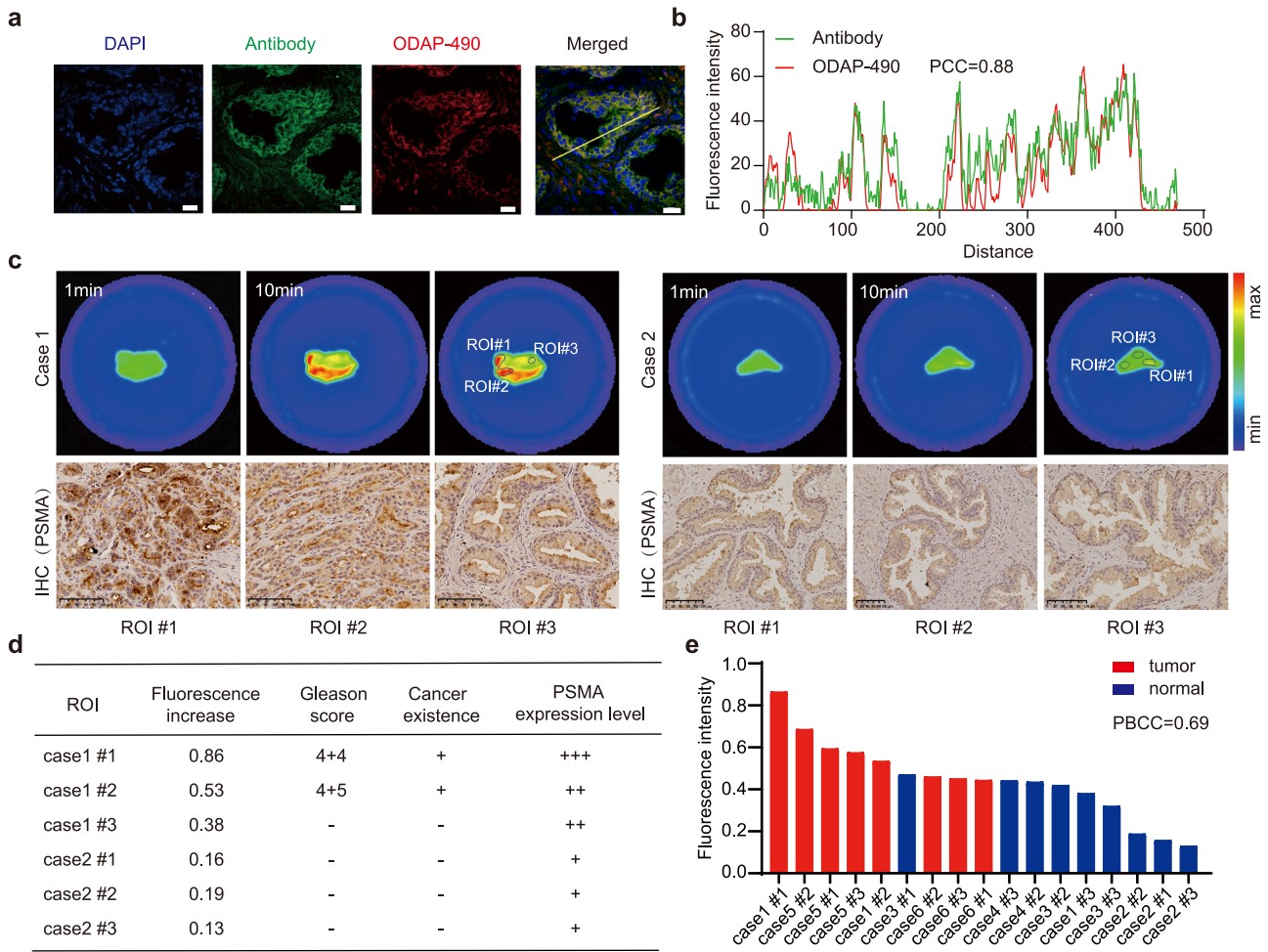

**Fig. 4 Wash-free imaging of PCa surgical specimens. a** Fluorescence image of surgical specimens co-stained by PSMA antibody and ODAP-490. Nucleus were stained with DAPI and shown in blue. Antibody and ODAP-490 staining were shown in green and red, respectively. Scale bar, 10 μm. Experiment was repeated three times independently with similar results. **b** Correlation between antibody and ODAP-490 stains. Fluorescence intensities corresponding to cross-section (yellow lines) in panel **a**. **c** Fluorescence image of surgical specimens incubating with ODAP-490. Representative images of corresponding PSMA immunohistochemistry (IHC) of in different regions of interest (ROIs) are shown below. Scale bar: 100 μm. IHC results were repeated three times independently with similar results. **d** Correlation of the fluorescence intensity, Gleason score, cancer existence, and PSMA expression level of each ROIs in panel **c**. **e** Summary of fluorescence increase at ROIs in six clinical specimens incubated with ODAP-490. PBCC point-biserial correlation coefficient. Source data are provided in Source Data file.

added to the mixture drop by drop with continuous stirring. The reaction mixture was stirred on ice bath for another 2 h. After the reaction completed, the mixture was diluted with 100 mL DCM and washed sequentially with 1 M NaOH solution twice, 10% CuSO$_4$ solution (w/w), brine and dried over Na$_2$SO$_4$. The crude product was purified using flash column chromatography to yield a colorless oil (4.0 g, yield 74%).

*Tert-butyl-2-cyano-3-(1-(3-methoxy-3-oxopropyl)-1,2,3,4-tetrahydroquinolin-6-yl) acrylate (6)*. Compound 5 (1.5 g, 6.07 mmol), *tert*-butyl cyanoacetate (4.4 mL, 30.3 mmol), pyridine (30 mL), and piperidine (1.0 mL) were added to a 250 mL round bottom flask. The reaction mixture was stirred under 90 °C for 8 h. After the reaction completed, the mixture was diluted with 100 mL ethyl acetate and washed sequentially with 1 N HCl twice, brine and dried over Na$_2$SO$_4$. The product was purified using flash column chromatography to yield a yellow solid (2.0 g, yield 89%).

*(4 S,8 S)-15-cyano-16-(1-(3-methoxy-3-oxopropyl)-1,2,3,4-tetrahydroquinolin-6-yl)-1,6,14-trioxo-2,5,7,13-tetraazahexadec-15-ene-1,4,8-tricarboxylic acid (ODAP-436)*. Compound 6 (0.9 g, 2.43 mmol) was dissolved in 10 mL DCM/TFA (v/v = 1:1). The reaction was stirred at room temperature for 2 h. After the *t*-butyl protection was removed, the solvent was removed under vacuum to provide a yellow oil. The yellow oil (176.5 mg, 0.56) was dissolved in 30 mL DCM, HATU (254.5 mg, 0.67 mmol) and DIPEA (762 μL, 2.8 mmol) were added into the reaction. The reaction was stirred at room temperature for 10 min, and compound 2b was added. The mixture was stirred for another 2 h. After the reaction completed, the solvent

was removed under vacuum. The crude material was purified using flash column chromatography to yield a yellow solid (284 mg). The yellow solid (80 mg) was dissolved in 10 mL DCM/TFA (v/v = 1:1) and the reaction was stirred for 2 h. After removing the solvent, the product was purified by reverse phase HPLC (0–3 min, 10% MeCN(0.1%TFA); 3–18 min, 10–70% MeCN(0.1%TFA); 18–25 min, 90% MeCN(0.1%TFA); Rt = 16.2 min) to yield ODAP-436 as a red solid (25.5 mg, yield 27%).

**Determination the fluorescence spectrum and property of probes**. Fluorescence and UV-Vis spectra of probes were recorded using a fluorescence spectro-meter (F-7000, Hitachi, Japan) and an ultraviolet-visible (UV-Vis) spectrophotometer (UH5300, Hitachi, Japan), respectively. UV-Vis spectra were acquired from 350 to 900 nm (0.2 nm increment). The fluorescence measurement of Glu-490 and ODAP-490 (0.1 mM) in PBS/glycerol (v/v = 1/1) solution were carried out at an excitation wavelength of 490 nm and emission spectra scan in the range of 500–900 nm, ODAP-436 were excited at the wavelength 436 nm and emission spectra scan in the range of 440–700 nm. For fluorescent enhancement property determination, probes were diluted in PBS-glycerol mixture with different glycerol fractions (0, 20, 40, 60, or 80% glycerol, v/v) to a final concentration of 0.1 mM. The fluorescence intensity was measured using a microplate reader (BioTek, Winooski, VT, USA) with λ$_{EX}$/λ$_{EM}$ = 436/495 nm for ODAP-436, λ$_{EX}$/λ$_{EM}$ = 490/660 nm for Glu-490 and ODAP-490. The fluorescence intensity in 0% glycerol was normalized to be 1 and each experiment was performed in three replicates.

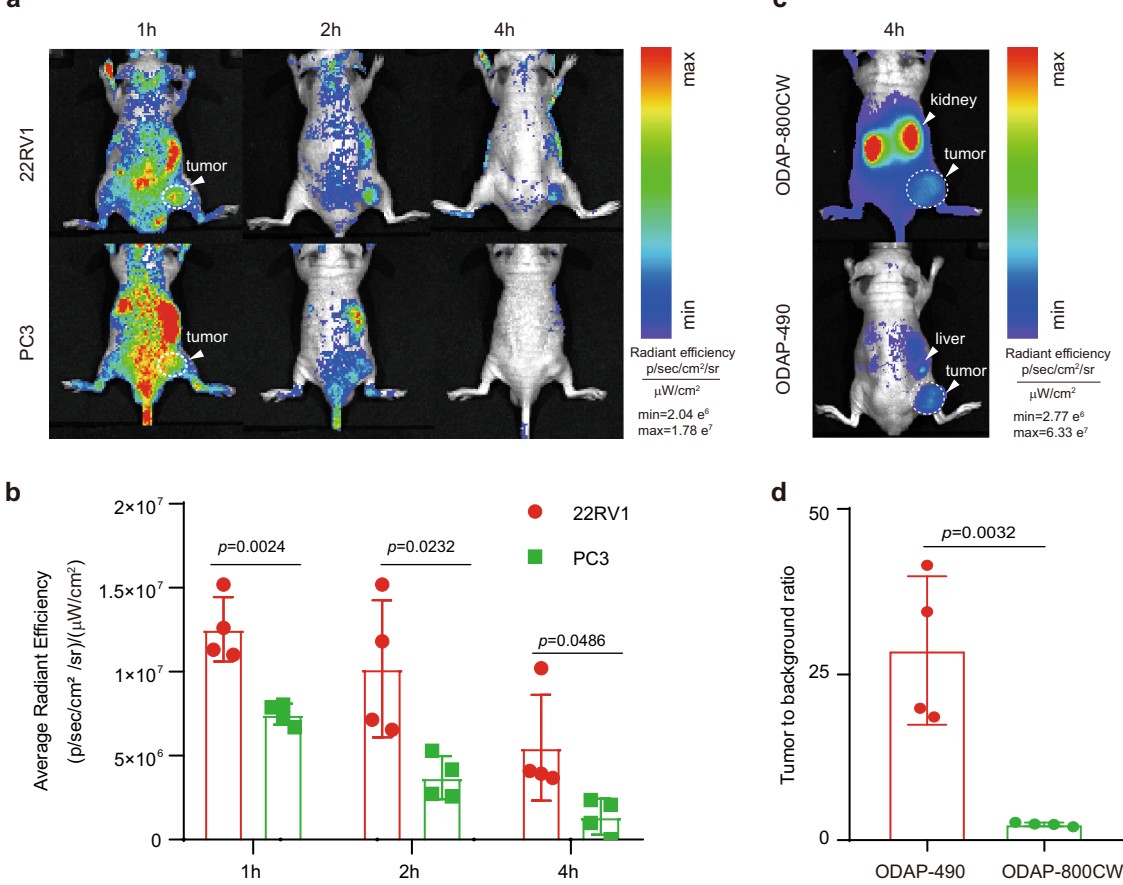

**Fig. 5 In vivo imaging in mouse model of prostate tumor. a** Fluorescence imaging of 22RV1 and PC3 tumors. Mice were intravenous injected with ODAP-490 and images were acquired at 1 h, 2 h and 4 h. The white arrow and dotted circle indicate the location of tumor. **b** Quantitative analysis of fluorescence intensity of 22RV1 and PC3 tumors in panel **a**. Data are presented as mean ± s.d. (*n* = 4 biologically independent mice). Two-tailed unpaired Student *t*-test. **c** Fluorescence imaging of PSMA 22RV1 tumor at 4 h after injected with ODAP-490 or ODAP-800CW. The white arrow in the upper and lower panels indicates kidney and liver, respectively. The white arrow and dotted circle indicate the location of tumor. **d** Quantitative analysis of tumor-to-background ratio of 22RV1 tumors in panel **c**. The tumor area was delineated with bright field and the same size of area was delineated on the opposite side as background. Data are presented as mean ± s.d. (*n* = 4 biologically independent mice). Two-tailed unpaired Student *t*-test. Source data are provided in Source Data file.

**rhPSMA expression and purification.** Expression and purification of the extracellular domain of human PSMA (rhPSMA; amino acids 44–750) were carried out with an established protocol[40]. Briefly, rhPSMA with a TEC-cleavable N-terminal Strep-tag was overexpressed in Schneider's S2 cells and concentrated to 1/10 of the original volume using tangential flow filtration (Millipore Mosheim, France). Concentrated medium was dialyzed against 50 mM Tris-HCl, 150 mM NaCl, pH 8.0, at 4 °C overnight and the fusion protein affinity purified using Streptactin Sepharose (IBA, Germany). The N-terminal Strep-tag was removed by the TEV protease (1:50 molar ratio) at 4 °C overnight and the final purification step included size exclusion chromatography on the Superdex 200 16/60 column (GE Healthcare Bio-Sciences, Uppsala, Sweden) in a running buffer comprising 20 mM Tris-HCl, 150 mM NaCl, pH 8.0. Purified rhPSMA (>98% purity as determined by SDS-PAGE) was concentrated to 10 mg/mL and kept at −80 °C until further use.

**Crystallization and data collection.** The stock solution of rhPSMA (10 mg/ml) was mixed with a 1/9 volume of 25 mM Glu-490 in 20 mM Tris-HCl and 150 mM NaCl at pH 8.0. The protein/inhibitor complex (1.5 µl) was mixed with 1 µl of the reservoir solution comprising 34% (v/v) pentaerythritol propoxylate PO/OH 5/4 (Sigma–Aldrich), 1% (w/v) PEG3350, and 100 mM Tris-HCl at pH 8.0. Crystals were grown in hanging drops using the vapor diffusion method at 293 K. For data collection, crystals were vitrified in liquid nitrogen directly from crystallization droplets. Diffraction data were collected from a single crystal at 100 K using synchrotron radiation at the MX14.2 beamline (0.92 Å; BESSYII, Helmholtz-Zentrum Berlin, Berlin, Germany). Datasets were indexed, integrated, and scaled using the XDSAPP interface. The data collection statistics are shown in Table 1.

**Structure determination and refinement.** Structure determination of the rhPSMA/Glu-490 complex was carried out using difference Fourier methods with the ligand-free hGCPII (Supplementary Note. 1, PDB code 7BFZ) as a starting model. Calculations were performed with the program Refmac 5.8., and the refinement protocol was interspersed with manual corrections to the model employing the program Coot 0.9. The restrains library and the coordinate file of the inhibitor were prepared using the ACEDRG, and the inhibitor was fitted into the positive electron density map in the final stages of the refinement. Approximately 2% of the randomly selected reflections were kept aside for cross-validation ($R_{free}$) during the refinement process. The quality of the final model was evaluated using the MOLPROBITY software and relevant statistics are summarized in Table 1.

**Fluorescence signal linearity of rhPSMA/probe complexes.** Two-fold dilution series of Glu-490, ODAP-490, or ODAP-436 (the final concentration range of 2 nM–1 µM) were mixed with a 10-fold molar excess of rhPSMA in the assay buffer comprising 50 mM Tris-HCl, 150 mM NaCl, 0.001% dodecyloctaglycol ($C_{12}E_8$) at pH 7.4 in a total volume of 20 µL in 384-well low volume flat-bottom black polystyrene microplates (Corning). The assay buffer was used as control. Samples were preincubated at room temperature for 10 min and fluorescence intensity was measured using a CLARIOstar microplate reader (BMG Labtech, MA, USA) with $\lambda_{EX}/\lambda_{EM}$ = 483/660 nm Glu-490 and ODAP-490, $\lambda_{EX}/\lambda_{EM}$ = 440/495 nm for ODAP-436. Data were fitted using a linear regression equation calculated in GraphPad Prism (San Diego, CA, USA). All reactions were carried out in triplicates and the data are shown as mean ± S.D.

**Saturation binding of rhPSMA/probe complexes.** Saturation experiments were carried out in the assay buffer using a constant concentration of 500 nM compounds and increasing concentrations of rhPSMA (two-fold dilution series; concentration range 8 nM–4 µM) in a total volume of 20 µL in 384-well low volume

flat-bottom black polystyrene microplates (Corning). Samples were preincubated at room temperature for 10 min and fluorescence intensity measured using a CLARIOstar microplate reader with $\lambda_{EX}/\lambda_{EM} = 483/660$ nm for Glu-490 and ODAP-490, $\lambda_{EX}/\lambda_{EM} = 440/495$ nm for ODAP-436. Data were fitted using a nonlinear regression analysis calculated in GraphPad Prism. All reactions were carried out in triplicates and the data are shown as mean ± S.D.

**Timeline of rhPSMA/ probe complex formation.** Five hundred nanomolar of Glu-490, ODAP-490, or ODAP-436 was mixed with two-fold excess of rhPSMA in the assay buffer in 384-well low volume flat-bottom black polystyrene microplates to a total volume of 20 μL. Immediately upon mixing the plate was inserted into a CLARIOstar microplate reader and the fluorescence intensity measured $\lambda_{EX}/\lambda_{EM} = 483/660$ nm for Glu-490 and ODAP-490, $\lambda_{EX}/\lambda_{EM} = 440/495$ nm for ODAP-436. All reactions were carried out in triplicates and the data are shown as mean ± S.D.

**Inhibition constants determination.** Inhibition constants were determined using the radioenzymatic assay with $^3$H-NAAG, radiolabeled at the terminal glutamate as a substrate as described previously[40]. Briefly, rhPSMA (2.5 ng/ml) was pre-incubated in the presence of increasing concentrations of inhibitors in 20 mM Tris and 150 mM NaCl at pH 8.0 for 15 min at 37 °C with a total volume of 80 μl. Reactions were initiated by addition of 40 μl of the mixture of cold NAAG (0.31 μM, Sigma) and $^3$H-NAAG (15 nM, 50 Ci/mmol, Perkin Elmer), and terminated after 4 h by the addition of 120 μl of 200 mM potassium phosphate, 50 mM EDTA, 2 mM β-mercaptoethanol at pH 7.4. The released glutamate was separated from the reaction mixture by ion-exchange chromatography and quantified by liquid scintillation. Duplicate reactions were carried out for each experimental point. The data were fitted using the GraphPad Prism software and $IC_{50}$ values were calculated from the inhibition curves of two independent experiments using a nonlinear analysis protocol.

**Cell lines and mouse models.** LNCaP, 22RV1, and PC3 human PCa cell lines were purchased from the Chinese Academy of Sciences Typical Culture Collection (Shanghai, China). LNCaP cells were grown in RPMI 1640 medium containing 10% fetal bovine serum (FBS), 1% penicillin–streptomycin, 1% GlutaMax-I, and 1% sodium pyruvate. 22RV1 and PC3 cells were grown in RPMI 1640 medium containing 10% fetal bovine serum (FBS), 1% penicillin–streptomycin. The cells were cultured at 37 °C under 5% $CO_2$ in air. All animal experiments were performed in accordance with ethical regulations on laboratory animals of the Beijing municipality. All procedures and protocols were approved by the Animal Ethics Committee at Peking University Frist Hospital (Beijing, China), approval number: J201987. BALB/c nude mice were obtained from the Animal Center at the Peking University Frist Hospital. Mice were group-housed (up to five mice in one cage), maintained in a 20–25 °C and humidity-controlled room with 12 h light/dark cycle. Before further experiments, all mice were acclimatized for at least 7 days. Four-week-old male, BALB/c nu mice were implanted subcutaneously with 22RV1 or PC3($10^7$ cells/mouse) cells on the back, respectively. When the xenografts reached 500-1000 mm³ mice were used for imaging.

**In vitro cytotoxicity.** PC3, 22RV1 and LNCaP cells were seeded in a 96-well culture plate at 5000 cells per well and incubated for 24 h. Cells were then treated with medium containing two-fold dilution series of ODAP-490 (concentration range 1.5625–50 μM) for 48 h. Cell viability was determined by the thiazolyl blue tetrazolium bromide (MTT) assay using a microplate reader (BioTek, Winooski, VT, USA). The relative viability of the untreated controls was normalized to be 100%, while the medium absorbance set as the background control. Each experiment was performed in triplicate.

**Wash-free staining of prostate cell lines.** PC3, 22RV1, and LNCaP cells were seeded in glass bottom culture dishes (NEST, San Diego, CA, USA) and cultured under 5% $CO_2$ atmosphere at 37 °C for 2 days. The cells were washed twice by PBS buffer and then incubated in no-phenol 1640 medium containing 10 μM ODAP-490 probes at 37 °C for 2 h. The images were captured using LSM880 confocal microscope (Zeiss, Germany) and Zen 2010 software. Image quantification was performed in ImageJ software (NIH).

**Real-time imaging of PSMA-mediated internalization.** LNCaP cells were seeded in an eight-well cell chamber and cultured under 5% $CO_2$ atmosphere at 37 °C for 2 days. The culture medium was replaced with fresh no-phenol 1640 medium containing 10 μM ODAP-490 and the fluorescence images were captured immediately using a confocal microscope equipped with a $CO_2$ incubator every 3 min for 2 h. To quantify the fluorescence intensity of cells, the single cell in the scope was delineated and the fluorescence intensity in this area was measured. Meanwhile, an area of the same size was delineated in the blank region and the fluorescence intensity in it was set as background. For inhibition experiments, before acquiring fluorescence images, LNCaP cells were pretreated with 10 μM endocytosis inhibitor chlorpromazine[50] or 100 μM PSMA inhibitor ZJ-43[49] for 30 min.

**PSMA immunofluorescence of frozen section of resected specimens.** A step-by-step protocol of immunofluorescence staining can be found at Protocol Exchange[57]. Fresh resected specimens were embedded in optimal cutting temperature compound (OCT) and stored at −80 °C until for immunofluorescence. The embedded specimens were cut into 7 μm-thick slides and fixed with cold acetone for 20 min. After being blocked by BSA for 1 h, the slides were incubated with a primary antibody against PSMA (ab19071; Abcam, Cambridge, UK; 1:1000 dilution) at 4 °C overnight, and then stained with FITC-labeled secondary antibody (ZF-0312; ZSGB-Bio, Beijing, China; 1:100 dilution) at room temperature for 1 h. Finally, Hochest 33342 was used to stain nuclei and a 10 μM ODAP-490 solution was applied before the slides were sealed. The FluoView1000 confocal microscope (Olympus, Japan) was used to scan the stained slides.

**Wash-free imaging of fresh resected PCa.** This study was approved by the Medical Ethics Committee of Peking University First Hospital (#2015-977). We have obtained informed consent from all participants before the experiments. Resected specimens were soaked in a 10 μM solution of ODAP-490 for 10 min at room temperature and the fluorescence images were immediately captured with a IVIS Spectrum Imaging System (Caliper life Sciences, Hopkinton, MA) with excitation at 490 nm and emission at 660 nm.

**PSMA immunohistochemistry of resected specimens.** Fixed specimens were embedded by paraffin and cut into 4 μm-thick slides. Antigen retrieval was applied at 100 °C at pH 9 for 20 min, and then the sample was washed three times with 1× PBS (pH 7.4). The primary antibody against PSMA (ab19071; Abcam, Cambridge, UK; 1:1000 dilution) was added to the slides and incubated overnight at 4 °C. A goat anti-mouse IgG (PV6002; ZSGB-Bio, Beijing, China) was used according to the handbook of manufacturer. 3,3′-Diaminobenzidine (DAB) was used as a chromogen, and hematoxylin was applied for counterstaining. Stained slides were scanned by Nano Zoomer-SQ pathological section scanner (Hamamatsu, Japan). The SPSS 27.0 software was used for correlation analysis between fluorescence increase and tumor existence.

**Penetration depth of ODAP-490.** We chose a scaffold of 1% intralipid as a simulated tissue for its similar scattering characteristics[58]. It was prepared by diluting 30% intralipid (Fresenius Kabi) with deionized water. A glass capillary tube filled with 10 μM ODAP-490 (diluted with 90% glycerol) was encapsulated for imaging. The capillary tube was then placed in a cylindrical culture dish and immersed in different volumes of 1% intralipid. The images were acquired using an IVIS imaging system with Ex/Em at 500/660 nm. Meanwhile, an area of the same size was delineated in the blank region, and the fluorescence intensity in it was set as background. The background signal was subtracted from the signal of the region of interest. Each experiment was performed in triplicate.

**In vivo imaging of mice bearing 22RV1 and PC3 prostate tumors.** PSMA + 22RV1 and PSMA-PC3 tumors were induced on mice by subcutaneous injection of $5 \times 10^6$ cells in suspension in 100 μL PBS buffer, respectively. The xenografts were used for in vivo imaging when their size reached approximately 0.5–1 cm³. For imaging of ODAP-490, tumor-bearing mice were administrated with 25 nmol ODAP-490 in 100 μL PBS buffer, and then the images were acquired at 1, 2, and 4 h by the IVIS Spectrum Imaging System (Caliper life Sciences, Hopkinton, MA). A spectral unmixing processing was used to subtract the autofluorescence, which was performed using Living Image 4.3.1 software (Caliper life Sciences, Hopkinton, MA). A series of images were acquired using auto-exposure with the following parameters: Ex/Em pairs: 500/640 nm, 500/660 nm, 500/680, 500/700 nm, 500/720 nm. For the in vivo imaging of ODAP-800CW, a series of images of tumor-bearing mice were acquired at 4 h post-injection of 1 nmol ODAP-800CW in 100 μL PBS buffer, with the same settings as ODAP-490 except following parameters: Ex/Em pairs: 710/760 nm, 710/780 nm, 710/800 nm, 710/820 nm, 710/840 nm, 745/800 nm, 745/820 nm, 745/840 nm. The tumor area was delineated with bright field and the same size of area was delineated on the opposite side as background.

**Statistical analysis.** All data are presented as means ± SD. Student's t-test was used to determine significance with Graphpad prism 8.0. Point-biserial correlation coefficient was calculated with SPSS 27.0 software. Statistical significance was considered at $P < 0.05$.

**Reporting summary.** Further information on research design is available in the Nature Research Reporting Summary linked to this article.

## Data availability

All data generated or analysed during this study are included in this published article (and its supplementary information files). The coordinates and structure factors data generated in this study have been deposited in the Protein Data Bank (PDB) under the accession code 7BFZ. Source data are provided with this paper.

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

## Acknowledgements

We thank Barbora Havlinova and Petra Baranova for their excellent technical assistance and Lucia Motlova for help with crystallization experiments. This work was financially supported by the National Natural Science Foundation of China (21877004, 92059101), Clinical Medicine Plus X—Young Scholars Project of Peking University (PKU2020LCXQ029). Additionally, this work was in part supported by the CAS (RVO: 86652036), the Czech Science Foundation (18-04790 S, 19-22269Y) and the National Institutes of Health (R01 CA134675). We acknowledge the Helmholtz-Zentrum Berlin for the allocation of synchrotron radiation beamtime at the MX14.2 beamline and the support by the project CALIPSOplus (grant agreement 730872) from the EU Framework Programme for Research and Innovation HORIZON 2020, and CMS-Biocev ("Crystallization/Diffraction") supported by MEYS CR (LM2018127).

## Author contributions

X.Y. conceived and designed research. X.D. performed the chemical synthesis. C.B., A.R., Z.K., J.Z., and X.D. performed in vitro PSMA protein assay biochemical, inhibition, and fluorescent experiments. C.B. and A.R. crystallized, refined, and analyzed the rhPSMA/Glu-490 complex. J.Z., Q.Y., and Y.W. performed live cell imaging and measured the absorption and emission spectrum. X.L., P.W., and Y.X. collected and provided the prostate cancer samples. J.Z. performed MTT assay and histopathological staining. C.A.F., J.Z., and M.G.P designed and performed the mouse imaging. J.Z., X.D., C.B., Q.Y., N.Z., and X.Y. analyzed data. X.Y., J.Z., C.B., X.C., Y.F., Z.Y., and M.G.P. co-wrote the paper. All authors discussed the results and commented on the manuscript.

## Competing interests

The authors declare no competing interests.
