## [Peer Review File · Nature Communications]

A prostate-specific membrane antigen activated molecular rotor for real-time fluorescence imagingReviewers' Comments:

Reviewer #1:

Remarks to the Author:

In this manuscript Zhang et al. coupled a red emitting molecular rotor to prostate-specific membrane antigen targeting moiety to develop a fluorogenic probes to localize prostate cancer. Overall the work is interesting and experiments are well conducted. the properties of the developed probes were well characterized (fluorescence enhancement, internalization pathway, histological comparisons, suitable controls) and the crystallography study of Glu-490 is a real asset in this study. Furthermore, the application is interesting and showed a clear potential in fluorescence guided surgery. However, the novelty of the molecular rotor is not new as it was recently described by Hsu et al. in Nat. Chem. 2019 [10.1038/s41557-019-0217-x]. Despite this lack of chemical novelty, I would recommend major revision, and I would ask the authors to address the following points:

Introduction.

A more detailed introduction on fluorogenic molecular rotor and a deeper review on excising probes for prostate cancer would be necessary.

Results.

- « Benzonitrile and julolidine are well known FMRs ». By themselves benzonitrile and julolidine are not FMRs . This sentence should be replaced by: "FMR based on benzonitrile and julolidine moieties are well known."
 - At the end of the "Design and synthesis of PSMA-targeted activatable probes" paragraph, the authors should state why they targeted more water-soluble probes. Was that to limit the non-specific interactions and thus avoid « false positive » staining?
 - Regarding the chemical design of the probes, the presence of the ester function is unclear. Was it to stick as much as possible to the probe from Hsu et al. Nat. Chem. 2019 ? If yes why? This ester is prone to esterases hydrolysis that could change the staining properties over the time.
 - The arrow representing the rotation is not placed correctly, indeed there is no rotation of the double bond. The rotation occurs at the sigma bonds. This should be corrected.
 - The NMR spectra of the final compounds and new intermediates should be provided as proof of purity.
 - According to the NMR description in SI only one diastereoisomer was isolated for both probes. However, the structures in fig1b indicate that the authors did not identify the diastereoisomers (waved bonds). I recommend to unambiguously confirm the stereochemistry of the probes (Z or E). This can be performed using NOESY and /or ¹³C-¹H coupled NMR (see 10.1039/d0qm00872a).
- PS : In the next section, the crystallography showed that Glu-490 is isomer E. Consequently, the structure in Fig. 1b should be changed accordingly.
- Fig1f caption ODAO should be changed for ODAP, no ?

Mechanism of fluorescent activation upon probe-PSMA binding

- Line 123: In chemistry, nitril is an electron acceptor group and tetrahydroquinoline is electron donor, not the opposite.
- Line 154: For consistency, ZJ43 should be written the same way ZJ-43.
- Figure 5a and c. For clarity, the tumor site should be indicated with a region of interest.

In vivo imaging.

- Comparison with ODAP-800CW. Fig 5c is confusing as the tumor site is not indicated. Therefore one

could think that the tumor could be the intense red spots (kidneys?) The picture should be more detailed by the authors to avoid confusion and the red regions should be clearly assigned.

- Moreover, is it really fair to compare the signal/background ratio of two probes with very different properties? The acquisition channels were different (excitation sources and emission filters), and ODAP-490/800CW used concentration ratio was 25-fold. The conclusion of this experiments should take these considerations into account.

Reviewer #2:

Remarks to the Author:

PSMA is an excellent biomarker for prostate cancer targeting and has attracted worldwide attention recently for developing radiopharmaceuticals with extremely high sensitivity and specificity. Zhang et al. described a novel PSMA-activated fluorescent imaging probe for real-time imaging applications. Besides the high potency, these probes can achieve nearly instantaneous activation of over 30-fold of robust fluorescent signal upon PSMA-probe binding with a very smart design. The ultra-fast activation character of this probe enabled to monitor the PSMA-mediated endocytosis process in real-time for the first time. The clear understanding of this process is critical for the success of PSMA targeting peptide receptor radionuclide therapy (PRRT) and this method could potentially provide a great tool for related investigation. Other interesting applications, such as wash-free tumor sample staining and in vivo imaging, were also described and the results are convincing with the clear potential for translational research. The mechanism of the fluorescent activation was clearly demonstrated by resolving X-ray structure of PSMA-probe complex, which I believe may provide insightful information for developing receptor-activated fluorescent probes in general. Overall, it's a high quality research with good novelty and translational potential, and I would recommend for publication in Nature Communications. The minor comments are listed below.

- 1) "Fig 1a." (Schematic of probe binding and activation) should be to move it into the last paragraph of introduction, where the design of the probe was mentioned.
- 2) In the result section, Design and synthesis of PSMA-targeted activatable probes, water solubility difference between Glu-490 and ODAP-490 were mentioned. Please provide the logP or logD of the two compounds for comparison.
- 3) In the result section, Biochemical and biophysical characterization of probes, fluorescent titration in glycerol solution is a known method for FMR characterization, instead of a new invention. Please provide an appropriate reference.
- 4) In Figure 2, please include the PDB code in the figure legend and related discussion.
- 5) In the result section, Real-time monitoring of live cell receptor binding and endocytosis, "medium to low" should be revised to "medium-to-low".
- 6) In Figure 3a, there is a cell viability drop at high concentration of the probe (50uM), which seems correlate with PSMA expression level of the cell lines. Please comment on that. Is it a general property of PSMA inhibitors or just to the probes?
- 7) In the discussion section, for the cell imaging, please include another key reference discussing the endocytosis of PSMA ligand recently just published in Cancer Research. (Cytoplasmic localization of prostate-specific membrane antigen inhibitors may confer advantages for targeted cancer therapies. Cancer Res. 2021 Feb 23;canres.1624.2020. doi: 10.1158/0008-5472.CAN-20-1624.)
- 8) In the method section, Inhibition constants determination, a radioenzymatic assay was applied. Please comment on the difference between the radioenzymatic assay and fluorescence-based assay

which is also commonly used for PSMA inhibition study.

9) Please check and revise the Supplementary Figure 1. There's format issue in the scheme.

10) In Supplementary Figure 2, please include the PDB code in the figure legend.

11) In Supplementary Figure 3, what is the potency difference between EIPA, Nystain and CPZ and what's the reason for choosing 10 μ M endocytosis inhibitors for the study.

Reviewer #3:

Remarks to the Author:

The manuscript by Zhang and colleagues describes the development and preclinical evaluation of a fluorescent PSMA ligand that gets activated once bound and internalized by PSMA. PSMA is currently a widely investigated target in prostate cancer as it provides a specific target.

As I am a clinician (urologist) I cannot judge the chemical and preclinical experiments. The described findings, however, seem interesting and encouraging.

Still, I have several questions/comments/suggestions:

- Figure 5a: arrow and circle are missing?
- As the authors discuss intraoperative real-time imaging to detect positive margins and replace frozen section analysis it would be interesting to know more about the tissue penetration of the fluorescent dye (from the mouse experiments it seems that the fluorescence signal is able to penetrate several millimeters whereas a few cell layers would provide a surgical clean margin) and the amount of tumor cells (or PSMA molecules) that have to be present to obtain a signal above background
- Do the authors have an idea on the route of metabolism (hepatobiliary, urinary excretion?) or on stability in vivo that may influence background signal or tumor-to-background ratio over time?

Point-by-point response to reviewers

We would like to thank the reviewers for the insightful and constructive comments! We have revised the manuscript according to their advices, which should significantly improve the clarity and quality of our work. Below is a list of the point-by-point responses to the reviewer comments.

Comments from Reviewer 1:

In this manuscript Zhang et al. coupled a red emitting molecular rotor to prostate-specific membrane antigen targeting moiety to develop a fluorogenic probes to localize prostate cancer. Overall the work is interesting and experiments are well conducted. the properties of the developed probes were well characterized (fluorescence enhancement, internalization pathway, histological comparisons, suitable controls) and the crystallography study of Glu-490 is a real asset in this study. Furthermore, the application is interesting and showed a clear potential in fluorescence guided surgery. However, the novelty of the molecular rotor is not new as it was recently described by Hsu et al. in Nat. Chem. 2019 [10.1038/s41557-019-0217-x]. Despite this lack of chemical novelty, I would recommend major revision, and I would ask the authors to address the following points:

R1.1 - Introduction.

A more detailed introduction on fluorogenic molecular rotor and a deeper review on excising probes for prostate cancer would be necessary.

R-R1.1: We thank the reviewer for this great suggestion. A deeper introduction to fluorogenic molecular rotors and existing probes for prostate cancer has been added. We also added more relevant references to enable readers to follow progress in the field.

(i) For fluorogenic molecular rotors, we added the following description:

Fluorescent molecular rotors (FMRs) are a family of fluorophores sensitive to local microenvironment (e.g. polarity and viscosity)¹⁸⁻²⁰. Upon photoexcitation, the molecule can form a low-energy state, referred to as a twisted intramolecular charge transfer (TICT) state, so that the excitation energy can be dissipated accompanied by red-shifted emission or non-radiative relaxation. The strategy for applying FMRs is to restrict the formation of the TICT state, in which case specific fluorescence enhancement (usually quantum yield and fluorescent lifetime) can be obtained²¹. The fluorescence response of FMRs is more sensitive and faster compared with other on-off probes mediated by specific chemical reactions²², enabling real-time and in-situ detection. In this regard, some FMRs have been developed for sensing viscosity in the microenvironment, such as derivatives of julolidine (DCVJ, CCVJ)^{23,24}, meso-phenyl-substituted derivatives of BODIPY²⁵⁻²⁷, porphyrin derivatives²⁸ and merocyanines dyes²⁹. Increasingly, FMRs have been exploited for imaging biomolecules (protein, DNA, RNA, peptidoglycan) and biomolecular interactions³⁰⁻³⁸. (line73-84)

(ii) For existing probes for prostate cancer, the description was revised as:

To date optical probes targeting prostate cancer have largely been based on fluorescent dyes (Cy7, IRdye800CW, indocyanine green etc.) that are “always on”, requiring substantial time to achieve suitable image contrast¹⁴. PSMA-activatable fluorescent probes, which have an off-on response, would be desirable as they may have less background than fluorescent agents that are continuously fluorescent upon excitation, but only a few examples have been reported¹⁵⁻¹⁷. Kobayashi et al. carried out pioneering studies based on the quenching effect when a fluorescent dye (such as indocyanine green) was conjugated with an antibody or minibody, demonstrating a 30-fold increase in fluorescence upon activation and internalization of the agent by PSMA^{15,16}. Based on that activation effect, they could detect PSMA-positive tumors specifically at 6 h post-injection with minibodies conjugated with indocyanine green. However, the requirement of internalization for activation, and the long clearance times of monoclonal antibodies from the blood pool, prevent imaging shortly after

administration. Urano *et al.* recently reported a novel fluorogenic method using the carboxypeptidase activity of PSMA. The method could be utilized for *ex vivo* fluorescence imaging of PCa in surgically resected clinical specimens¹⁷. The method enabled high fluorescence enhancement, while the generation of signal depended on the enzymatic activity of PSMA, and requires minimum 30 mins for sample staining. Overall, existing methods need at least tens of minutes to hours to reach suitable image contrast, encouraging us to develop probes that may enable imaging at shorter times post-administration. (line54-72)

References added:

14. Chen, Y. *et al.* Synthesis and biological evaluation of low molecular weight fluorescent imaging agents for the prostate-specific membrane antigen. *Bioconjug. Chem.* **23**, 2377–2385 (2012).
18. Haidekker, M. A. & Theodorakis, E. A. Molecular rotors - Fluorescent biosensors for viscosity and flow. *Org. Biomol. Chem.* **5**, 1669–1678 (2007).
20. Lee, S. C. *et al.* Fluorescent molecular rotors for viscosity sensors. *Chem. Eur. J.* **24**, 13706–13718 (2018).
22. Liu, H. W. *et al.* Recent progresses in small-molecule enzymatic fluorescent probes for cancer imaging. *Chem. Soc. Rev.* **47**, 7140–7180 (2018).
24. Haidekker, M. A. *et al.* New fluorescent probes for the measurement of cell membrane viscosity. *Chem. Biol.* **8**, 123–131 (2001).
25. Kuimova, M. K., Yahioğlu, G., Levitt, J. A. & Suhling, K. Molecular rotor measures viscosity of live cells via fluorescence lifetime imaging. *J. Am. Chem. Soc.* **130**, 6672–6673 (2008).
26. Duarte, I. L., Vu, T. T., Izquierdo, M. A., Bull, J. A., & Kuimova, M. K. A molecular rotor for measuring viscosity in plasma membranes of live cells. *Chem. Commun.* **50**, 5282–5284 (2014).
27. Sherin, P. S. *et al.* Visualising the membrane viscosity of porcine eye lens cells using molecular rotors. *Chem. Sci.* **8**, 3523–3528 (2017).
28. Kuimova, M. K. *et al.* Imaging intracellular viscosity of a single cell during photoinduced cell death. *Nat. Chem.* **1**, 69–73 (2009).
29. Mukherjee, T. *et al.* Near infrared emitting molecular rotor based on merocyanine for probing the viscosity of cellular lipid environments. *Mater. Chem. Front.* **5**, 2459–2469 (2021).
30. Kummer, S. *et al.* Fluorescence imaging of influenza H1N1 mRNA in living infected cells using single-chromophore FIT-PNA. *Angew. Chem. Int. Ed.* **50**, 1931–1934 (2011).
31. Paige, J. S., Wu, K., & Jaffrey, S. R. RNA mimics of green fluorescent protein. *Science.* **333**, 642–646 (2011).
33. Dziuba, D., Jurkiewicz, P., Cebecauer, M., Hof, M. & Hocek, M. A rotational BODIPY nucleotide: An environment-sensitive fluorescence-lifetime probe for DNA interactions and applications in live-cell microscopy. *Angew. Chem. Int. Ed.* **55**, 174–178 (2016).
34. Karimi, A., Börner, R., Mata, G. & Luedtke, N. W. A highly fluorescent nucleobase molecular rotor. *J. Am. Chem. Soc.* **142**, 14422–14426 (2020).
35. Ye, S., Zhang, H., Fei, J., Wolstenholme, C. H. & Zhang, X. A general strategy to control viscosity sensitivity of molecular rotor-based fluorophores. *Angew. Chem. Int. Ed.* **60**, 1339–1346 (2021).
36. Telmer, C. A. *et al.* Rapid, specific, no-wash, far-red fluorogen activation in subcellular compartments by targeted fluorogen activating proteins. *ACS Chem. Biol.* **10**, 1239–1246 (2015).
38. Karpenko, I. A. *et al.* Push-pull dioxaborine as fluorescent molecular rotor: Far-red fluorogenic probe for ligand-receptor interactions. *J. Mater. Chem. C.* **4**, 3002–3009 (2016).

R1.2 -Results. « Benzonitrile and julolidine are well known FMRs ». By themselves benzonitrile and julolidine are not FMRs. This sentence should be replaced by: “FMR based on benzonitrile and julolidine moieties are well known.”

R-R1.2: We thank the reviewer's suggestion. The text has been revised as suggested.

R1.3 - At the end of the "Design and synthesis of PSMA-targeted activatable probes" paragraph, the authors should state why they targeted more water-soluble probes. Was that to limit the non-specific interactions and thus avoid « false positive » staining?

R-R1.3: We thank the reviewer for pointing this out. We hypothesized that more hydrophilic probes may have less non-specific interaction with the cell membrane, which is exactly what the reviewer has suggested. A brief description has been added in the manuscript as suggested:

*Additionally, we synthesized **ODAP-490** and **ODAP-436**, Lys-Urea-oxalyldiaminopropionic acid analogs of the traditional glutamate-containing scaffold, to increase its water solubility and reduce non-specific interaction. (Fig. 1b). (line100-102)*

*As **Glu-490** and **ODAP-490** were virtually indistinguishable in our biophysical experiments, and **ODAP-490** exhibited less non-specific staining than **Glu-490** (Supplementary Fig. 5), we selected **ODAP-490** for the ensuing biological applications. (line154-157)*

In addition, as the proof of this hypothesis, we selected LNCaP and PC3 cells as PSMA positive and negative cell lines, respectively, and stained them with either **Glu-490** or **ODAP-490**. The results showed that **ODAP-490** has the same staining properties for LNCaP cells and less non-specific staining for PC3 cells, compared with **Glu-490**. These data have been added to supplementary information (now Supplementary Fig. 8).

Supplementary Figure 8, Wash-free imaging of LNCaP and PC3 using Glu-490 and ODAP-490. (a) Confocal imaging of LNCaP and PC3 stained with **Glu-490** and **ODAP-490**. White arrows indicate the nonspecific staining. Scale bar: 20 μm . **(b)** Quantification of the fluorescence intensity of LNCaP cells in panel **a**, ($n = 30$ biologically independent cell samples). **(c)** Quantification of the fluorescence intensity of PC3 cells in panel **a**, ($n = 30$ biologically independent cell samples). MFI: Mean Fluorescence Intensity. Two-tailed unpaired Student *t*-test.

R1.4 - Regarding the chemical design of the probes, the presence of the ester function is unclear. Was it to stick as much as possible to the probe from Hsu et al. Nat. Chem. 2019? If yes why? This ester is prone to esterases hydrolysis that could change the staining properties over the time.

R-R1.4 We thank the reviewer for pointing it out. The ester (**ODAP-490**) could be synthesized more

efficiently than the carboxylic acid compound (**ODAP-490-COOH**), which required one-step hydrolysis with moderate yield and further purification by HPLC (~22% yield after purification). We prepared and characterized both **ODAP-490** and **ODAP-490-COOH** (see below), and they showed very similar properties when binding to PSMA. For the stability, **ODAP-490** is stable in human serum albumin (HSA) making it suitable for *in vitro* or *ex vivo* staining, but in murine serum it is slowly hydrolyzed to **ODAP-490-COOH** as the only stable product, with similar *in vivo* imaging characteristics. Considering the similar functional characteristics of both compounds, we chose to investigate **ODAP-490**, which could be scaled up more efficiently. All the proofs, including 1) the synthesis of **ODAP-490-COOH**, 2) the fluorescence characterization of **ODAP-490-COOH**, 3) stability test of **ODAP-490**, were listed below and provided in SI.

The synthesis of **ODAP-490-COOH** was listed below and all the related spectra were also provided in SI.

(4S,8S)-16-(5-(1-(2-carboxyethyl)-1,2,3,4-tetrahydroquinolin-6-yl)thiophen-2-yl)-15-cyano-1,6,14-trioxo-2,5,7,13-tetraazahexadec-15-ene-1,4,8-tricarboxylic acid

ODAP-490 (12 mg, 0.017 mmol) was dissolved in THF/H₂O (v/v=1:1, 2mL), then LiOH (7mg, 0.17 mmol) was added and the reaction mixture was stirred for 15min at room temperature. Then, the mixture was neutralized to pH = 7 with 1M HCl and concentrated under reduced pressure. The crude product was purified by HPLC (0-5 min, 5% MeCN (0.1%TFA); 5-15 min, 10%-45% MeCN (0.1%TFA); 15-25 min, 45% MeCN (0.1%TFA); 25-26 min, 45%-90% MeCN (0.1%TFA); 26-31 min, 90% MeCN (0.1%TFA) to yield as a red solid (3mg, yield 22%).

The functional characteristics of **ODAP-490-COOH** were assessed as for **ODAP-490**. Biochemical experiments revealed virtual equivalence of the two compounds with similar response to viscosity changes, picomolar inhibitory potency, strong fluorescence enhancement, fast activation time and long signal stability.

Supplementary Figure 5. Functional characteristics of ODAP-490-COOH. (a) Structure of **ODAP-490-COOH**. (b) Changes in fluorescence intensity of **ODAP-490-COOH** in solutions with increasing glycerol concentrations. Probe concentration was 0.01 mM and data were normalized to PBS control. Mean \pm s.d. ($n=3$ biologically independent experiments) (c) Inhibition of PSMA enzymatic activity using the radioenzymatic assay. Mean \pm s.d. ($n=2$ biologically independent experiments) (d) Saturation binding of rhPSMA/probe complexes. ($n=2$ biologically independent experiments) (e) Fluorescence intensity of the rhPSMA/probe complex in response to concentration changes. ($n=2$ biologically independent experiments) (f) Time frame for rhPSMA/probe complex formation. Mean \pm s.d. ($n=3$ biologically independent experiments).

In vitro stability of **ODAP-490** was determined by incubation in a PBS buffer containing 10% HSA or murine serum. We found that **ODAP-490** is stable in human serum albumin (HSA) for 12 hours, and there was 69% **ODAP-490** left after incubating with mice serum for 4 hours, and **ODAP-490-COOH** was the only hydrolyzed product observed. In addition, *in vivo* experiments (see R3.3) further showed nearly identical tissue biodistribution. Overall, both biochemically and *in vivo* property of the carboxylate derivative are similar to **ODAP-490**.

Supplementary Figure 10. In vitro stability of ODAP-490. (a) HPLC of ODAP-490 incubated with human serum albumin (HSA) for different times. (b) HPLC of ODAP-490 incubated with murine serum.

R1.5 - The arrow representing the rotation is not placed correctly, indeed there is no rotation of the double bond. The rotation occurs at the sigma bonds. This should be corrected.

R-R1.5: We appreciate the reviewer for pointing it out. The position of the arrow has been moved to the sigma bond as suggested (Fig. 1a). In addition, the structure has been modified following the crystallography as suggested by **R-R1.8**.

R1.6 - The NMR spectra of the final compounds and new intermediates should be provided as proof of purity.

R-R1.6: The NMR spectra of the final compounds and intermediates have been added to the supplementary information as the proof of purity (Supplementary Fig. 3).

R1.7 - According to the NMR description in SI only one diastereoisomer was isolated for both probes. However, the structures in fig1b indicate that the authors did not identify the diastereoisomers (waved bonds). I recommend to unambiguously confirm the stereochemistry of the probes (Z or E). This can be performed using NOESY and /or ^{13}C - ^1H coupled NMR (see 10.1039/d0qm00872a).

R-R1.7: We thank this reviewer for taking note and suggesting ways to address this issue. We have double checked the ^1H -NMR of all the related structures (all spectra have been added into SI as **R-R1.6**) and they show a single isomer, in agreement with what has been described by Hsu et al. in Nat. Chem. 2019 [10.1038/s41557-019-0217-x]. Furthermore, we obtained a NOESY spectrum for **Glu-490** (below). We found a weak NOE interaction between the proton of the alkene (in red, at 8.27 ppm, singlet) and N-H (in blue, 8.24 ppm, triplet) in DMSO (d_6), in agreement with the X-ray co-crystal structure, supporting the E isomer. Because in the ^1H -NMR the chemical shifts are very close and overlapping (8.27 ppm vs 8.24 ppm), we believe that the X-ray structure provides more conclusive proof for the E isomer. We have revised the structures in Fig.1b as the E isomer.

R1.8 - PS: In the next section, the crystallography showed that Glu-490 is isomer E. Consequently, the structure in Fig. 1b should be changed accordingly.

R-R1.8: We thank the reviewer for pointing this out. All structures in Fig.1b have been changed to the E isomer, the same conformer as observed in the X-ray structure.

R1.9 - Fig1f caption ODAO should be changed for ODAP, no?

R-R1.9: We appreciate the reviewer for pointing it out. Sorry for the typo and we have revised it as "ODAP".

R1.10 - Mechanism of fluorescent activation upon probe-PSMA binding

- Line 123: In chemistry, nitril is an electron acceptor group and tetrahydroquinoline is electron donor, not the opposite.

R-R1.10: We thank the reviewer for pointing this out. We have revised it as:

The FMR moiety consists of a distal N-substituted tetrahydroquinoline donor group, a thiophene ring spacer and a nitrile acceptor group. (line139-140)

R1.11 - Line 154: For consistency, ZJ43 should be written the same way ZJ-43.

R-R1.11: We thank the reviewer for this suggestion. We have revised all “ZJ43” to “ZJ-43” for consistency.

R1.12 - Figure 5a and c. For clarity, the tumor site should be indicated with a region of interest.

R-R1.12: Thanks for pointing it out. We have marked the site of tumor with a white dotted circle and arrow as suggested.

R1.13 - In vivo imaging.

- Comparison with ODAP-800CW. Fig 5c is confusing as the tumor site is not indicated. Therefore one could think that the tumor could be the intense red spots (kidneys?) The picture should be more detailed by the authors to avoid confusion and the red regions should be clearly assigned.

R-R1.13: We appreciate the reviewer’s suggestions. We have indicated the tumor site with a white dotted circle and arrow. Meanwhile, the intense red areas in Figure 5c have been designated as kidney and liver, respectively.

R1.14- Moreover, is it really fair to compare the signal/background ratio of two probes with very different properties? The acquisition channels were different (excitation sources and emission filters), and ODAP-490/800CW used concentration ratio was 25-fold. The conclusion of this experiments should take these considerations into account.

R-R1.14: We thank the reviewer for the comment. The optical properties of these two probes are different, so we optimized the concentration effect to reach the maximum signal-to-background ratio for both probes and applied the optimum concentration for the experiment. We found 25 nmol injection dose for **ODAP-490** could reach higher contrast than 10 nmol or 50 nmol, and 1 nmol injection dose for **ODAP-800CW** could reach higher contrast than 10 nmol and 25 nmol. We have listed these additional data into the supporting information (Supplementary Fig. 13). We hope this data can clarify the reviewer’s concern.

Supplementary Figure 13. In vivo imaging in mouse model of prostate tumor with different doses of probes. (a) Fluorescence imaging of 22RV1 tumors after injection with different doses of ODAP-490. Mice were intravenously injected with 10 nmol, 25 nmol or 50 nmol ODAP-490 and images were acquired at 1h, 2h and 4h. The white arrow and dotted circle indicate the location of tumor. (b) Quantitative analysis of tumor to background ratio of 22RV1 tumors in panel a. Mean \pm s.d. ($n = 4$ for 25 nmol group, $n = 3$ for 10 nmol and 50 nmol group), two-tailed unpaired Student *t*-test. (c) Fluorescence imaging of 22RV1 tumors after injection with different doses of ODAP-800CW. Mice were intravenously injected with 1 nmol, 10 nmol or 25 nmol ODAP-800CW and images were acquired at 1h, 2h and 4h. The white arrow and dotted circle indicate the location of tumor. (d) Quantitative analysis of tumor to background ratio of 22RV1 tumors in panel c. Mean \pm s.d. ($n = 4$ for 1 nmol group, $n = 3$ for 10 nmol and 25 nmol group), two-tailed unpaired Student *t*-test.

Comments from Reviewer 2:

PSMA is an excellent biomarker for prostate cancer targeting and has attracted worldwide attention recently for developing radiopharmaceuticals with extremely high sensitivity and specificity. Zhang et. al. described a novel PSMA-activated fluorescent imaging probe for real-time imaging applications. Besides the high potency, these probes can achieve nearly instantaneous activation of over 30-fold of robust fluorescent signal upon PSMA-probe binding with a very smart design. The ultra-fast activation character of this probe enabled to monitor the PSMA-mediated endocytosis process in real-time for the first time. The clear understanding of this process is critical for the success of PSMA targeting peptide receptor radionuclide therapy (PRRT) and this method could potentially provide a great tool for related investigation. Other interesting applications, such as wash-free tumor sample staining and in vivo imaging, were also described and the results are convincing with the clear potential for translational research. The mechanism of the fluorescent activation was clearly demonstrated by resolving X-ray structure of PSMA-probe complex, which I believe may provide insightful information for developing receptor-activated fluorescent probes in general. Overall, it's a high quality

research with good novelty and translational potential, and I would recommend for publication in Nature Communications. The minor comments are listed below.

R2.1 - “Fig 1a.” (Schematic of probe binding and activation) should be to move it into the last paragraph of introduction, where the design of the probe was mentioned.

R-R2.1: We thank the reviewer for the suggestion. We have moved “Fig 1a.” to the last paragraph of introduction.

R2.2 - In the result section, design and synthesis of PSMA-targeted activatable probes, water solubility difference between Glu-490 and ODAP-490 were mentioned. Please provide the logP or logD of the two compounds for comparison.

R-R2.2: We thank the reviewer for the suggestion. We have calculated the logD of **Glu-490** and **ODAP-490** using ACD/Labs (ACD/Labs, version 6.0; Advanced Chemistry Development: Toronto, Canada), and their clogD values (pH 7.40) were -3.96 and -4.05, respectively. That suggested that the Lys-Urea-oxalyldiaminopropionic acid scaffold has higher water solubility. In addition, we have determined the water solubility difference between the Lys-Urea-Glu scaffold and the Lys-Urea-ODAP scaffold by measuring the logP value of radioisotope labeled analogs of ^{68}Ga -PSMA617 and ^{68}Ga -7. The logP value of ^{68}Ga -PSMA617 and 7 were -2.00 ± 0.27 and -2.62 ± 0.30 , respectively. Those data have been added to the supporting information (Supplementary Fig. 4).

Supplementary Figure 4. Water solubility difference between Lys-Urea-Glu scaffold and Lys-Urea-ODAP scaffold. (a) The logD values of **ODAP-490** and **Glu-490** were calculated by ACD/Labs. (b) Structures of ^{68}Ga -PSMA617 and ^{68}Ga -7, with the experimental logP values of -2.00 ± 0.27 and -2.62 ± 0.30 , respectively.

R2.3 - In the result section, Biochemical and biophysical characterization of probes, fluorescent titration in glycerol solution is a known method for FMR characterization, instead of a new invention. Please provide an appropriate reference.

R-R2.3: We thank the reviewer for pointing it out. The description has been revised and a key reference has been added, as:

To evaluate fluorescence enhancement of our FMR probes in relation to their molecular environment, initially, we used a glycerol solution to mimic rotationally constrained conditions likely found upon PSMA binding and collected fluorescence and UV-Vis spectra of the compounds³⁷.” (line103-106)

The experiments were performed as in *Nat. Chem.* 2019, 11, 335–341.

R2.4 - In Figure 2, please include the PDB code in the figure legend and related discussion.

R-R2.4: Thanks. We have added the PDB code (7BFZ) to the figure legend and related discussion.

R2.5 - In the result section, Real-time monitoring of live cell receptor binding and endocytosis, “medium to low” should be revised to “medium-to-low”.

R-R2.5: Thanks. We have revised this to “medium-to-low”.

R2.6 - In Figure 3a, there is a cell viability drop at high concentration of the probe (50uM), which seems correlate with PSMA expression level of the cell lines. Please comment on that. Is it a general property of PSMA inhibitors or just to the probes?

R-R2.6: We thank the reviewer for this comment. To address this concern, we have performed a cell viability study on a Glu-based inhibitor (ZJ-43, *J. Neurochem.* 2004, 89, 876–885.), an ODAP-based inhibitor, (ODAP-FITC, *J. Med. Chem.* 2020, 63, 3563–3576.) and ODAP-490 with a FMR conjugated to the ODAP-based inhibitor, for comparison. We found that, unlike ODAP-490, the PSMA inhibitor ZJ-43 and ODAP-FITC did not show significant cytotoxicity at high concentration. That result suggests that the cytotoxicity of ODAP-490 is a property unique to this compound at relative high concentration, which seems related with the FMR part and PSMA mediated endocytosis. We have put these additional data into supporting information (Supplementary Fig. 7).

Supplementary Figure 7, Cytotoxicity comparison of ZJ-43, ODAP-FITC and ODAP-490 determined by the MTT viability assay. (a-c) Structures of ZJ-43, ODAP-FITC and ODAP-490. (d-f) Cytotoxicity of ZJ-43, ODAP-FITC and ODAP-490 at different concentrations. (g-i) Comparison of cytotoxicity between 0 μM and 50 μM for each compound. Data were shown as mean ± s.d.

R2.7 - In the discussion section, for the cell imaging, please include another key reference discussing the endocytosis of PSMA ligand recently just published in Cancer Research. (Cytoplasmic localization of prostate-specific membrane antigen inhibitors may confer advantages for targeted cancer therapies. *Cancer Res.* 2021 Feb 23; canres.1624.2020. doi: 10.1158/0008-5472.CAN-20-1624.)

R-R2.7: We appreciate the reviewer's suggestion. The work of Matthias et. al. published recently revealed the subcellular fate of PSMA/inhibitors complex, which is an important complement to the study of PSMA internalization. We have included it into the discussion section, as:

That is consistent with the established results that most transmembrane receptors are internalized via a clathrin-dependent mechanism, as recently shown by Matthias et al., who did so using stimulated emission depletion (STED) nanoscopy⁵⁶. (line230-232)

R2.8 - In the method section, Inhibition constants determination, a radioenzymatic assay was applied. Please comment on the difference between the radioenzymatic assay and fluorescence-based assay which is also commonly used for PSMA inhibition study.

R-R2.8: Thanks. Radioenzymatic assay with ³H-NAAG as a substrate is regularly used in the field by us and others (*Anal Biochem.* 2002, 310, 50-54.). Other assays, including a fluorescence-based assay using a coupled enzymatic relay to detect released glutamate (Amplex red assay, *J. Med. Chem.* 2020, 63, 3563-3576), are also used by some laboratories. From the biochemical perspective, these assays are interchangeable as they both quantify released glutamate from the NAAG substrate. Understandably, several basic requirements shall be considered such as: (i) substrate concentrations are the same and below the Km value; (ii) PSMA concentration is in a lower picomolar range to allow for measurement of such high affinity interactions; (iii) studied inhibitors do not interfere with the coupled reactions and/or the fluorescence readout and/or are not oxidized by hydrogen peroxide formed during the generation of fluorescence. Of note, several years ago we compared both assays side-by-side and did not find any substantial differences between the two (unpublished results).

R2.9 - Please check and revise the Supplementary Figure 1. There's format issue in the scheme.

R-R2.9: We appreciate the reviewer for pointing this out. We have revised the format of Supplementary Figure 1.

R2.10 - In Supplementary Figure 2, please include the PDB code in the figure legend.

R-R2.10: Thanks. We have added the PDB code (7BFZ) to the figure legend.

R2.11 - In Supplementary Figure 3, what is the potency difference between EIPA, Nystain and CPZ and what's the reason for choosing 10 μM endocytosis inhibitors for the study.

R-R2.11: We thank the reviewer for this comment. EIPA, Nystain and CPZ are commonly used endocytosis inhibitor which can inhibit the macropinocytotic, caveolin-dependent and clathrin-dependent pathways of endocytosis, respectively (*Methods Mol. Biol.* 2008, 440, 15–33.). All inhibitors inhibit endocytosis of their respective pathway(s) in the micromolar range. For cell culture experiments, the dose of inhibitor required to inhibit endocytosis may vary for different types of cells, with high doses being cytotoxic. The concentration of endocytosis inhibitor we used in this study was based on published work (*Mol. Pharm.* 2009, 6, 959-970.).

Comments from Reviewer 3:

The manuscript by Zhang and colleagues describes the development and preclinical evaluation of a fluorescent PSMA ligand that gets activated once bound and internalized by PSMA. PSMA is currently a widely investigated target in prostate cancer as it provides a specific target.

As I am a clinician (urologist) I cannot judge the chemical and preclinical experiments. The described findings, however, seem interesting and encouraging.

Still, I have several questions/comments/suggestions:

R3.1 - Figure 5a: arrow and circle are missing?

R-R3.1: We appreciate the reviewer for pointing this out. We have added the white dotted circle and arrow

in Fig 5a.

R3.2 - As the authors discuss intraoperative real-time imaging to detect positive margins and replace frozen section analysis it would be interesting to know more about the tissue penetration of the fluorescent dye (from the mouse experiments it seems that the fluorescence signal is able to penetrate several millimeters whereas a few cell layers would provide a surgical clean margin) and the amount of tumor cells (or PSMA molecules) that have to be present to obtain a signal above background.

R-R3.2: We appreciate the reviewer's great suggestions. Tissue penetration of fluorescent probes is an important factor that affects the tumor signal we can collect. In order to obtain the tissue penetration of the fluorescence probe after activated by PSMA, we measured the depth-of-penetration following an established procedure (*J. Am. Chem. Soc.* 2019, 141, 19221-19225.) as follows:

We chose a scaffold of 1% intralipid as a simulated tissue for its similar scattering characteristics. It was prepared by diluting 30% intralipid with deionized water. A glass capillary tube filled with 10 μM ODAP-490 (diluted with 90% glycerol) was encapsulated for imaging. The capillary tube was then placed in a cylindrical culture dish and immersed in different volumes of 1% intralipid. The images were acquired using an IVIS imaging system with Ex/Em at 500/660 nm. Meanwhile, an area of the same size was delineated in the blank region, and the fluorescence intensity in it was set as background. The background signal was subtracted from the signal of the region of interest. Each experiment was performed in triplicate. (line449-456)

For the convenience of the analysis of detection limit, LNCaP cells were used for the calculation. The mean number of PSMA binding site on LNCaP cells is 5.9×10^5 per cell (*J. Nucl. Med.* 2015, 56, 1401-1407.), which indicates the concentration of the PSMA is around $1.87 \mu\text{M}$. And we found that the penetration depth at this concentration is 4 mm. The penetration depth decreases as the concentration decreases. We hope this information could solve the reviewer's concern on the detection limit, which is the combination effect of receptor density and depth. This additional data has been included into supporting information.

Supplementary Figure 11. Penetration depth of ODAP-490. (a) Fluorescence images of ODAP-490 immersed in different depth of 1% intralipid. (b) Signal to background ratio in panel a. The background signal was subtracted from the signal of the region of interest. Mean \pm s.d. ($n = 3$ biologically independent experiments).

R3.3 - Do the authors have an idea on the route of metabolism (hepatobiliary, urinary excretion?) or on stability *in vivo* that may influence background signal or tumor-to-background ratio over time?

R-R3.3: Thanks for the reviewer's comments. To address the concerns, we performed stability and bio-distribution studies. For *in vivo* stability, we proved the methyl ester remote from the key molecular structure is the only unstable functional group, which could be hydrolyzed *in vivo* but would not affect the fluorescent properties of the molecule or the interaction with PSMA (**R-R1.4** for details). To determine the route of metabolism of the probes, we have tested the bio-distribution of **ODAP-490** at different times after injection (Supplementary Fig. 12a, b). We found that this probe was mainly metabolized within the liver, with less excreted by the kidneys. The hydrolyzed product of **ODAP-490** was also tested, and the results showed that both compounds have very similar pharmacokinetic properties (Supplementary Fig. 12c, d). Those additional data have been included into supporting information.

Supplementary Figure 12, Bio-distribution of ODAP-490 and metabolites. (a) The image of the excised organs before injection and 1 h, 4 h, and 10 h post-injection of 25 nmol **ODAP-490**. The organs sampled include: (1) liver, (2) spleen, (3) lung, (4) kidneys, (5) stomach, (6) bone, (7) heart, (8) muscle, (9) intestine. (b) Quantification of the fluorescence intensity of organs in panel a, Mean \pm s.d. ($n = 3$ biologically independent mice for each time point). (c) Image of the excised organs before injection and 1 h, 4 h, and 10 h post-injection of 25 nmol hydrolyzed **ODAP-490**. (d) Quantification of the fluorescence intensity of organs in panel c, Mean \pm s.d. ($n = 3$ biologically independent mice for each time point).

Reviewers' Comments:

Reviewer #1:

Remarks to the Author:

The authors addressed my comments (and other reviewer's ones) in a very conscientious manner. I therefore recommend to accept the revised manuscript for publication.

Reviewer #2:

Remarks to the Author:

I think the authors have adequately addressed all my comments.

Point-by-point response to reviewers

We would like to thank the reviewers for the insightful and constructive comments! Below is a list of the point-by-point responses to the reviewer comments.

Comments from reviewer #1 (Remarks to the Author):

The authors addressed my comments (and other reviewer's ones) in a very conscientious manner. I therefore recommend to accept the revised manuscript for publication.

We thank the reviewer for the very supportive comments.

Comments from reviewer #2 (Remarks to the Author):

I think the authors have adequately addressed all my comments.

We thank the reviewer for the very supportive comments.